

# A technology-based global non-methane volatile organic compounds (NMVOC) emission inventory under the MEIC framework

Ruochong Xu[1], Hanchen Ma[1], Jingxian Li[2], Dan Tong[1], Liu Yan[2], Lanyuan Wang[2], Xinying Qin[1], Qingyang Xiao[2], Xizhe Yan[1], Hanwen Hu[2], Yujia Fu[2], Nana Wu[1], Huaxuan Wang[2], Yuexuanzi Wang[1], Xiaodong Liu[2], Guannan Geng[2], Kebin He[2,3], and Qiang Zhang[1]

[1]Department of Earth System Science, Ministry of Education Key Laboratory for Earth System Modeling, Institute for Global Change Studies, Tsinghua University, Beijing 100084, People's Republic of China

[2]State Key Joint Laboratory of Environment Simulation and Pollution Control, School of Environment, Tsinghua University, Beijing 100084, People's Republic of China

[3]Institute for Carbon Neutrality, Tsinghua University, Beijing 100084, People's Republic of China

*Correspondence to*: Qiang Zhang (qiangzhang@tsinghua.edu.cn)

**Abstract.** Non-methane volatile organic compounds (NMVOC) play a crucial role in tropospheric atmospheric chemistry. Developing accurate NMVOC emission inventories is essential for chemical transport modeling and pollution mitigation. While global NMVOC emissions have been estimated in several inventories, representing the impacts of technology evolution on NMVOC emission dynamics remains challenging. Here, we developed a technology-based global NMVOC emission inventory (MEIC-global-NMVOC) under the Multi-resolution Emission Inventory model for Climate and air pollution research (MEIC) framework to better represent NMVOC emission evolution and drivers. We compiled spatially and temporally full-coverage and consistent activity rates from multiple sources using data fusion and reconstruction approaches. Next, we estimated the evolution of technology distributions and emission control measures by data-driven and policy-driven technology turnover models. The dynamics of global anthropogenic NMVOC emissions during 1970-2020 were presented by sector, fuel type, and product, highlighting activity growth and technology development as key drivers. In developing regions, emission growths were driven by surges in activity rates but curbed by advanced combustion technologies, improved vehicle standards, and substitution of solvent-borne paints. In developed regions, besides the mitigation measures above, emission control technologies substantially reduced fugitive emissions from energy production and emissions from solvent use, driven by policies such as the U.S. New Source Performance Standards and European Union Directives. Despite large uncertainties, MEIC-global-NMVOC emission estimates were generally comparable with other emission inventories at aggregated level for global or regional total, but remarkable sectoral differences remained. The MEIC-global-NMVOC emission inventory offers a new database for atmospheric chemistry and pollution mitigation research.

## 1. Introduction

Non-methane volatile organic compounds (NMVOC) encompass all organic compounds with a saturation vapor concentration $(C^*)$ exceeding $3\times10^6$ μg/m$^3$, excluding methane (Bianchi et al., 2019). NMVOC play an important role in tropospheric chemistry as precursors to both ozone ($O_3$) and secondary organic aerosol (SOA) (Hallquist et al., 2009; Shrivastava et al., 2017; Heald and Kroll, 2020). A portion of NMVOC is also classified as hazardous air pollutants (HAPs) (Strum and Scheffe, 2016). Thus, NMVOC have substantial global impacts on air quality, climate, and human health (Fry et al., 2014; Laurent and Hauschild, 2014; Li et al., 2019). Previous studies have demonstrated that mitigating anthropogenic NMVOC emissions is the key to controlling $O_3$ and fine particulate matter (PM$_{2.5}$) pollution, as well as addressing other negative effects in various regions (West et al., 2007; Fry et al., 2014; Sharma et al., 2016; Hendriks et al., 2016; Li et al., 2019). Therefore, accurately representing global anthropogenic NMVOC emissions is crucial for understanding their impacts on atmospheric chemistry and for designing effective pollution mitigation strategies worldwide.



Several global NMVOC emission inventories have been developed based on bottom-up estimates or data integration approaches (Huang et al., 2017; Crippa et al., 2018; Hoesly et al., 2018; Meng et al., 2019). These emission inventories consistently present the increasing trend of global NMVOC emissions, and reflect the emission magnitudes derived from in-situ observations (von Schneidemesser et al., 2023). However, the discrepancies of global NMVOC emissions can exceed 10% among the inventories (e.g., a global difference exceeding 15 Mt in 2018 between EDGAR version 6.1 and CEDS version 2021-04-21, two widely-used inventories; Crippa et al., 2018; Hoesly et al., 2018), and larger gaps can be found for major emitters and specific sectors (Li et al., 2019; Sharma et al., 2015; Kurokawa and Ohara, 2020), highlighting the large uncertainties in current NMVOC emission estimates. On the one hand, these discrepancies arise from differences in activity rates. Unlike carbon dioxide ($CO_2$) and nitrogen oxides ($NO_x$), which are mainly emitted from fuel combustion with high-quality energy statistics, the estimates of NMVOC emissions—especially from solvent use and industrial process—rely on various production and consumption data, which are often of poor quality or missing even for large economies (UNSD, 2022a). Different methods are used to integrate multiple data or fill the gaps, but are generally not presented in detail. On the other hand, incomplete or inconsistent considerations of emission-related technologies, such as vehicle standards, solvent types, and end-of-pipe control measures, further contribute to divergent emission estimates (Li et al., 2017).

Therefore, a few important limitations in current global NMVOC emission inventories should be concerned. First, there is a lack of detailed approaches to harmonize global activity rates from various sources with different quality and coverage, particularly for solvent use and chemical industry, leading to opacity in emission estimates and hard-to-assess differences. Second, although the evolution of technology penetration and emission control is generally considered for air pollutants like $SO_2$ and $PM_{2.5}$ (Crippa et al., 2018), it is rarely accounted for in major NMVOC-emitting sectors, resulting in potential biases (Mo et al., 2021). It is still challenging to represent the evolution of NMVOC-emission-related technologies across sectors with different emission characteristics and countries with different mitigation efforts. Third, there is a lack of comparison among global NMVOC emission inventories and their validation with observations. von Schneidemesser et al. (2023) compared global NMVOC emission inventories with observations and highlighted the need for further improvements in the representation and accuracy of current inventories. This calls for more efforts in developing global NMVOC emission inventories to provide new independent data sources.

Here, we developed a new technology-based global NMVOC emission inventory under the bottom-up framework of the Multi-resolution Emission Inventory model for Climate and air pollution research (MEIC) (Xu et al., 2024), to better represent the evolution and drivers of NMVOC-emission-related technologies and global NMVOC emissions. We first developed data fusion and reconstruction approaches to obtain spatially and temporally full-coverage and consistent activity rates from various data sources, particularly for reconstructing the activity rates for chemical industry and solvent use. We then modeled the evolution of technology distributions and emission control measures using data-driven and policy-driven technology turnover models. Global anthropogenic NMVOC emissions from 1970 to 2020 were then estimated using emission factors obtained from international databases, local literatures, and emission models. Based on the new inventory, we tracked the dynamics of global NMVOC emissions across sectors, fuel types, and specific products, presented the region-specific features of emission structures and evolution, and revealed the driving factors of emission changes. Finally, we compared our estimates with other global and regional inventories, providing uncertainty analysis and discussing the limitations.

## 2. Methods and data

### 2.1 Source categories

A detailed five-level source category system was developed to represent major anthropogenic NMVOC emission sources and associated technologies, extended upon our previous work on global $CO_2$ emission inventory (i.e., MEIC-global-$CO_2$ inventory;



Xu et al., 2024). The source category definitions are presented in Table S1 and described here in order from broad to specific. The top level consists of six major sectors which conform to the category classification in the IPCC 2006 Guidelines (IPCC GL): energy consumption for power and industry (referred to as energy-power and industry; IPCC GL category codes 1A1 and 1A2), energy consumption for commercial and residential (energy-commercial and residential; the majority of IPCC GL category code 1A4), energy consumption for transport (energy-transport; IPCC GL category code 1A3 and a subset of 1A4), fugitive emissions from energy (energy-fugitive emissions; IPCC GL category code 1B), industrial process (IPCC GL category codes 2A, 2B, 2C, 2H, and a subset of 2D), and solvent use (the majority of IPCC GL category code 2D).

The second level includes 51 combustion-related, 9 process-related, and 3 solvent-use sectors, subdivided from the major sectors. These sectors are also compatible with IPCC GL definitions, but are refined beyond the IPCC GL to better represent sector-specific emissions (Xu et al., 2024). The third level encompasses fuel types, industrial products, and processes used or produced in different sectors, which include 42 fossil fuels, 9 biofuels, 11 fugitive energy-related processes, 46 industrial products, and 15 solvent-consuming processes, all in alignment with IPCC GL classification.

At the fourth level, we considered technologies related to fuel combustion, industrial production, and solvent use. For the transport sector, emission standards are differentiated by vehicle category. In this study, vehicle emission standards across countries are mapped to the European standards for consistency in the source categories. The fifth level includes emission control technologies, accounting for the penetration ratios of various control measures with different removal efficiencies. Note that open biomass burning, agriculture, and waste treatment are not included in this work.

## 2.2 Definition of countries and territories

MEIC-global-NMVOC includes 228 countries/territories, an extension from the 208 countries/territories in the MEIC-global-$CO_2$ inventory (Xu et al., 2024). The additional 20 countries are included because the population data used as the activity rates of a few solvent-consuming sources cover more small countries than the energy statistics used in MEIC-global-$CO_2$. The detailed list and their mapping to the ISO-3166 country codes are presented in Table S2.

## 2.3 Model framework

NMVOC emissions by country from 1970 to 2020 were estimated under the bottom-up framework of the MEIC model (Xu et al., 2024). The emissions were calculated using a technology-based methodology, extended from our previous work in China (Zheng et al., 2018; Li et al., 2019). Briefly, emissions from stationary non-point sources were estimated following equation (1):

$$E_i = \sum_j \sum_k A_{i,j,k} \times \left[ \sum_m X_{i,j,k,m} \times EF_{j,k,m} \right] \quad (1)$$

where $i$ and $j$ represent country and sector, respectively. $k$ represents the fuel type for combustion-related sources or the product for process-related sources and fugitive emissions, and $m$ represents the technology used for fuel combustion or industrial production as well as the type of solvent consumed. $E$ represents the NMVOC emissions, calculated based on activity rate ($A$), technology distribution ($X$), and emission factor ($EF$). The emission factor was determined by integrating unabated emission factor ($EF_{unabated}$), penetration ratios of control technologies ($C_n$), and corresponding removal efficiencies ($\eta_n$), as shown in equation (2):

$$EF = EF_{unabated} \times \sum_n [C_n \times (1 - \eta_n)] \quad (2)$$

For thermal power, iron and steel, and cement industries, NMVOC emissions were estimated as point source using facility-




level data from the Global Infrastructure emission Detector (GID, http://gidmodel.org.cn/) (Qin et al., 2022; Chen et al., 2022; Xu et al., 2023), as shown in equation (3).

$$E_i = \sum_p A_{i,j,p} \times EF_{i,j,p} = \sum_p A_{i,j,p} \times EF_{unabated,i,j,p} \times (1 - \eta_{i,j,p}) \quad (3)$$

where $p$ represents facility, and the meanings of other symbols are the same as those in equations (1) and (2). GID includes capacity, utilization rates, fuel types, combustion or production technologies, and control technologies for 101,607 thermal
power units, 3,234 iron and steel plants, and 4,196 cement plants globally. The facility-level emissions were aggregated by country for analysis in this study.

NMVOC emissions from non-stationary sources (i.e., transport) were estimated using a fleet-based approach, which modeled vehicle population, fleet turnover rates, and implementation of emission standards by vehicle category (Zheng et al., 2014; Yan et al., 2024), as summarized in equations (4) and (5):

$$E_i = \sum_v \sum_f \sum_s \sum_a (Stock_{i,v} \times FuelR_{i,v,f} \times VKT_{i,v,f} \times FE_{i,v,f} \times X_{i,v,f,s} \times Age_{i,v,f,s,a} \times EF_{i,v,f,s,a}) \quad (4)$$

$$EF_{i,v,f,s,a} = EF_{running,i,v,f,s,a} + EF_{start,i,v,f,s,a} + EF_{evap,i,v,f,s,a} \quad (5)$$

where $v$ and $f$ represent vehicle category and fuel type, respectively. $s$ represents emission standard, and $a$ represents vehicle age. NMVOC emissions were calculated based on vehicle ownership ($Stock$), shares of fuel types ($FuelR$), annual average mileage ($VKT$), fuel economy ($FE$), distributions of emission standards ($X$) and vehicle ages ($Age$), and emission
factor ($EF$). The emission factor includes three emission modes: running ($EF_{running}$), start ($EF_{start}$), and evaporation ($EF_{evap}$). Details of the fleet-based approach are described in Sect. 2.4.3.

The technology-based approach used in this study represents the evolution of activity rates, technology distributions, unabated emission factors, and emission control, as summarized in Table 1. Emission control was further represented by dynamically changing penetration ratios and constant removal efficiencies of different control technologies. Specifically, for the energy-
transport sector, emission factors were provided by fleet-based turnover models that integrated unabated emission factors and emission control synchronously, and thus penetration of control technology and removal efficiency were not estimated separately. For the energy-power and industry sector, since fuel combustion is highly complete, NMVOC emission control is rarely implemented and was not considered here. For the energy-commercial and residential sector, fuel combustion in stoves generally lacks end-of-pipe control, so emission control was also not considered. The detailed methodology for deriving the
parameters is described in Sect. 2.4.

**Table 1.** Emission parameters considered in this study.

| Source category | Activity rates | Technology distributions | Unabated emission factors | Penetration of control technology | Removal efficiency |
|---|---|---|---|---|---|
| Energy- | | | | | |
|    Power and industry | √ | × | √ | × | × |
|    Commercial and residential | √ | √ | √ | × | × |
|    Transport | √ | √ | Combined* | | |
|    Fugitive | √ | × | √ | √ | √ |
| Industrial process | √ | × | √ | √ | √ |
| Solvent use | √ | √ | √ | √ | √ |

*The emission factors were provided by fleet-based models that integrated unabated emission factors and emission control synchronously.





### 2.4 Activity rates, technology distributions, and emission factors

In this section, we present the details of the technology-based approach to obtain activity rates, technology distributions, and emission factors. Estimates of global NMVOC emissions require a wide variety of activity rates for energy production and consumption, industrial production, and product use. Here, data fusion and reconstruction approaches were developed to process the activity rates, especially to fill the gaps in data for chemical production and solvent use, and finally generated complete and consistent time series of country-level activity rates from 1970 to 2020. A detailed list of source categories, along 155 with their sources and types of activity rates, is presented in Table S3.

The evolution of technology distributions and emission control measures was estimated by technology turnover approaches. For technology distributions, a data-driven technology turnover approach was applied. Technology penetration ratios related to fuel/product consumption and production were collected from literatures, research reports, and market surveys for available years, and then statistical gap-filling methods were used to reconstruct the full time series. For emission control, a policy-160 driven technology turnover model was used to estimate the penetration of NMVOC control technologies (Klimont et al., 2009; Lei et al., 2011). The implementation of environmental policies, legislation, and emission standards was considered as the driving force for the adoption and upgrading of control technologies. The penetration ratios of new control technologies ($C_{new}$) were estimated by equation (6):

$$C_{new,i,j,k,m,n,y} = \int_{t_0}^{y} S_{i,j,k,m,y} \quad t_0 \leq y < t_1 \quad (6)$$

where $y$ represents year, and $t_0$ and $t_1$ represent the start and end years of a policy/standard. The meanings of other symbols are the same as those in equations (1) and (2). $S$ represent technology penetration rate. The penetration ratios of existing control technologies ($C_{existing}$) were estimated by equation (7):

$$C_{existing,i,j,k,m,n,y} = \left(100\% - C_{new,i,j,k,m,n,y}\right) \times \frac{C_{existing,i,j,k,m,n,t0}}{\sum_n C_{existing,i,j,k,m,n,t0}} \quad t_0 \leq y < t_1 \quad (7)$$

In this way, the full time series of control technology evolution was represented. Emission factors were then calculated by 170 integrating penetration ratios of control technologies, removal efficiencies, and unabated emission factors obtained from first-hand measurements, technical guidebooks, and local surveys (see Table S4 and Supplementary Text S1). The data sources and detailed methodologies for deriving emission parameters are described by major sector as below.

### 2.4.1 Energy: power and industry

Activity rates of energy consumption in the power and industry sector were obtained following our previous work on global 175 $CO_2$ emission inventory (i.e., MEIC-global-$CO_2$; Xu et al., 2024). Briefly, a data-driven approach was developed to integrate data from the International Energy Agency (IEA), the Carbon Emission Accounts and Datasets (CEADs), and the Multi-resolution Emission Inventory for China (MEIC-China) (IEA, 2022; Cui et al., 2023; Zheng et al., 2018), generating seamless and highly-resolved data cubes of energy consumption in 1970-2021 for 208 countries/territories, 797 sub-country administrative divisions, 42 fuel types, and 52 sectors. Among the sectors, the energy-power and industry sector in this study 180 includes power generation, another 17 energy-producing industries, and 14 manufacturing industries as listed in Table S1. Detailed descriptions of the approach can be found in Sect. 2.4 and 2.5 of Xu et al. (2024) and are not presented here.

Since end-of-pipe NMVOC control measures are rarely implemented in the energy-power and industry sector, unabated emission factors were directly used as the final emission factors without considering control technologies. For power industry, unabated emission factors were obtained from the AP-42 database (US EPA, 1995; 2001). For manufacturing industries, 185 unabated emission factors were obtained from the EMEP Guidebook and literatures (EEA, 2023; Li et al., 2019; Klimont et





al., 2002). The differences of unabated emission factors were not distinguished among countries and specific industries.

### 2.4.2 Energy: commercial and residential

Activity rates of energy consumption in the commercial and residential sector were obtained in the same way as the power and industry sector. However, besides the fossil fuels included in MEIC-global-$CO_2$, biomass consumption data from IEA were also incorporated, including primary solid biofuel, biogas, charcoal, and other biofuels (See Table S1).

We developed a technology turnover model to derive the penetration of traditional and advanced stoves for residential combustion, reflecting the varying emission levels of different stove types. Data on stove type proportions were collected from literatures for 29 countries, which together accounted for approximately 74% of global residential energy consumption in 2020. After a series of modeling tests, the besting performing non-linear model was selected to fit the collected data, with GDP per capita and urbanization rate as independent variables. The model was verified by 10-fold cross-validation. It was then used to estimate the full time series of stove type evolution for all countries, representing the increasing adoption of advanced stove as economies grow and urbanize. More details on the model are presented in Supplementary Text S2.

Since fuel consumption in the commercial and residential sector generally lacks end-of-pipe control, unabated emission factors were directly used as the final emission factors. Fuel-type- and stove-type-dependent unabated emission factors were taken from the EMEP Guidebook as global default values (EEA, 2023), and further replaced with local measurements where available (e.g., Keita et al., 2021).

Figure S1 and Table S5 present the modeled proportions of advanced stove and average emission factors by region, weighted by residential energy consumption in each country. Developed regions, such as Canada and the U.S, had high proportions of advanced stoves. In some developing regions like China, the proportions of advanced stoves increased rapidly from 1970 to 2020, while in other regions, such as rest of Asia (including developing South and Southeast Asian countries) and rest of world (including African countries), the proportions remained low. Such technological gap notably influenced regional emission factors. For example, in China and India, average emission factors for biomass combustion declined by 27% and 19% in 1970-2020, respectively. In developed regions, average emission factors were generally lower and also decreased in 1970-2020, but the decreasing magnitude was smaller due to the already high penetration of advanced stove in 1970. Overall, the gap in average emission factors has narrowed among the regions as combustion technology improved over time.

### 2.4.3 Energy: transport

Activity rates of energy consumption in the energy-transport sector were also obtained following the approach used in MEIC-global-$CO_2$ (Xu et al., 2024). For road transport, a fleet-based approach was used to obtain the activity rates of different vehicle categories (i.e., car, bus, light-duty truck, heavy-duty truck, motorcycle, and other vehicles consuming LPG or other fuels), constrained by country-level fuel consumption from IEA (Yan et al., 2024). For off-road transport, activity rates covered aviation, navigation, railway, pipeline, machinery for agriculture, forestry, fishing, and other non-specified off-road vehicles.

In terms of technology distributions, we developed a fleet-based turnover model to derive the emission-standard-specific distributions within on-road vehicle fleets for each country, capturing the dynamic processes of fleet upgrading and emission mitigation. Based on the timing of emission standard implementation in each country, the model quantitatively represented the evolution of vehicle-category-, age-, and emission-standard-specific vehicle ownership by statistically modeling the fleet upgrading, aging, and phasing-out. Details of the model can be found in our previous studies (Zheng et al., 2014; Yan et al., 2024) and are not presented here.

For off-road transport, we also developed a simplified fleet-based turnover model to estimate the emission-standard-specific distributions. The model focused on off-road vehicles for agriculture, forestry, and fishing that consume diesel and gasoline,



excluding aviation, navigation, railway, and pipeline transport. For the EU, annual age structure and emission-standard-specific
       proportions within each age group of the fleets were directly obtained from the EMEP Guidebook (EEA, 2023). For other large
       economies (e.g., the U.S., China, and Japan), we compiled their emission standards for off-road vehicles from 1970 to 2020.
       We then applied the same proportions of new vehicles (i.e., 0-year-old vehicles) as in the EU when a new emission standard
       was implemented, assuming that all new vehicles complied with the latest emission standard to model fleet upgrading. For

countries without available emission standards, we assumed that all vehicles met the EU pre-1981 standard.

       Emission-standard-specific emission factors for on-road transport were calculated for start, running, and evaporation modes,
       which integrated unabated emission factors and emission control synchronously. Global default emission factors for each
       emission standard were obtained from the EMEP Guidebook (EEA, 2023), and further replaced with local data where available
       (e.g., Zheng et al., 2014). For start and evaporation modes, the effects of ambient temperature were considered in the calculation,

as higher temperatures lead to more NMVOC evaporation, while lower temperatures increase start-mode emission factors due
       to longer time required for the catalyst to reach operating conditions. For running emissions, vehicle deterioration was
       considered, which results in increased emission factors as cumulative mileage rises. For off-road transport, emission-standard-
       specific emission factors were obtained from the MOtor Vehicle Emission Simulator (MOVES) for North America and the
       EMEP Guidebook for other regions (US EPA, 2021; EEA, 2023). Detailed calculations are presented in Supplementary Text

S1.

       Figure 1 shows the modeled evolution of emission-standard-specific fleets for on-road gasoline, on-road diesel, and off-road
       diesel vehicles, as well as the corresponding changes in global fleet-average emission factors. The distributions clearly
       illustrate the global fleet upgrading by replacing outdated vehicles with higher-emission-standard ones. For example, in 1970,
       the global fleet of on-road gasoline vehicles consisted entirely of vehicles with the Pre-Euro I standard. By 2020, the proportion

of Pre-Euro I vehicles had decreased to ~10%, while vehicles with Euro V or higher standards dominated the fleet (~51%). A
       similar trend was observed for off-road vehicles. Regionally, fleet upgrading began earlier in developed regions compared to
       developing ones, though the process varied by vehicle category. By 2020, high-emission-standard vehicles were concentrated
       in EU27 and the U.K., Canada and the U.S., and China (Figs. S2 and S3).

       As the fleet upgraded with improved emission standards, the global fleet-average emission factors decreased by 77%, 65%,

and 55% in 1970-2020 for on-road gasoline, on-road diesel, and off-road diesel vehicles, respectively. The declining trends
       accelerated after 2000, driven by the rapid and extensive upgrading of large vehicle fleets in China (Figs. S2 and S3).



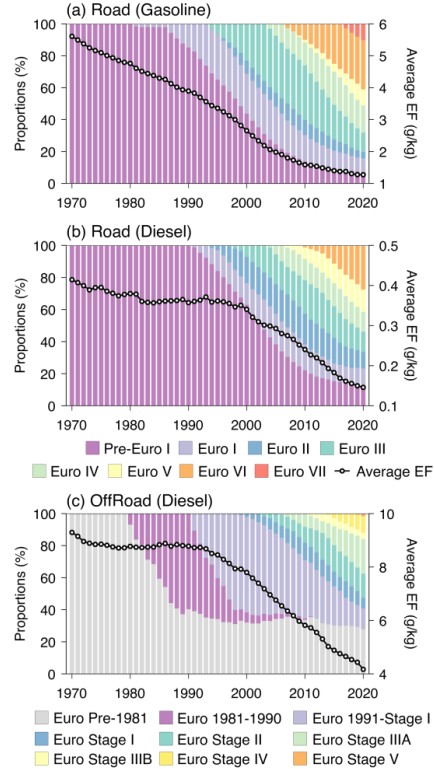

**Figure 1.** Proportions of emission-standard-specific (a) on-road gasoline vehicles, (b) on-road diesel vehicles, and (c) off-road diesel vehicles in the global fleet (colored bars), along with the changes in global fleet-average emission factors for start and running modes (black lines with scatters).

### 2.4.4 Energy: fugitive emissions

Activity rates of fugitive emissions were obtained from the IEA World Energy, Oil, and Natural Gas Statistics (IEA, 2022). The methodology followed our previous work on MEIC-global-$CO_2$ (Xu et al., 2024), but unlike energy consumption in the sections above, no regional data were integrated.

Unabated emission factors for oil- and gas-related fugitive emissions were taken from the EMEP Guidebook and the AP-42 database (EEA, 2023; US EPA, 1995; 2001). Unabated emission factors for coke production were obtained from literatures (Klimont et al., 2002; Bo et al., 2008). The differences of unabated emission factors were not distinguished among countries.

To estimate the penetration of control technologies, we first compiled a set of regional policies and legislation on NMVOC emission control. As NMVOC mitigation was mainly implemented in developed countries, policies/legislation were collected for the U.S. and EU. For Japan, the 2005 amendment to the Air Pollution Control Act was considered as the starting point for NMVOC control. For other developed countries, we mapped them to a country/region above based on geographical proximity (e.g., Canada was mapped to the U.S.; see Table S6). Since China began to control NMVOC emissions in the recent decade (Li et al., 2019; Geng et al., 2021), relevant policies and legislation in China were also compiled. No emission control was considered for other developing countries. The policy/legislation timelines for fugitive emissions are shown in Fig. 2a.





**(a) Energy-fugitive emissions**

**(b) Industrial process**

**(c) Solvent use**



**Figure 2.** Major policies and legislation on NMVOC emission control in the U.S., EU, and China for (a) energy-fugitive emissions, (b) industrial process, and (c) solvent use during 1970-2020. NSPS: New Source Performance Standards; NESHAP: National Emission

Standards for Hazardous Air Pollutants; LRTAP: Long-Range Transboundary Air Pollution; BREF: Best Available Techniques (BAT) reference document.

Next, control technologies and their removal efficiencies were obtained from technical guidebooks and measurements, including the EMEP Guidebook (EEA, 2023), EU BAT reference documents (BREFs) (e.g., EU-BRITE, 2007), US EPA procedures documents (US EPA, 2001), and literatures (Li et al., 2019). Control technologies for fugitive emissions include

vapor recovery system and leak detection and repair (LDAR) etc. with distinct removal efficiencies (See Table S7).

Then the adoption of control technologies for each policy/legislation was determined in accordance with the corresponding emission standards. If emission limits were specified (e.g., concentrations in end-of-pipe flue gases), the control technology that can reduce the uncontrolled emission level to the emission limit was selected. If no emission limit was provided, we selected a control technology widely used by regional producers during that period based on technical guidebooks and market

surveys (EEA, 2023; US EPA, 2001).

Finally, the penetration of control technologies was modeled as shown in equation (6) and (7). Technology penetration rates were estimated using S-shaped curves, following previous studies (Bond et al., 2007; Shen et al., 2013). The values of parameters used in S-shaped curves are presented in Table S8.

Figures 3a-3c present the normalized changes in emission factors for three large sources of energy-fugitive emissions across

Canada and the U.S., EU27 and the U.K., and China. Emission factors in all regions saw notable decreases from 1970 to 2020, but the reductions were more rapid and substantial in developed regions compared to China. For example, emission factors for oil refining in Canada and the U.S. decreased by approximately 70% in 2000 and 90% in 2020 relative to the 1970 level. In contrast, China only saw an obvious reduction after 2015, with a decrease of about 40% by 2020.

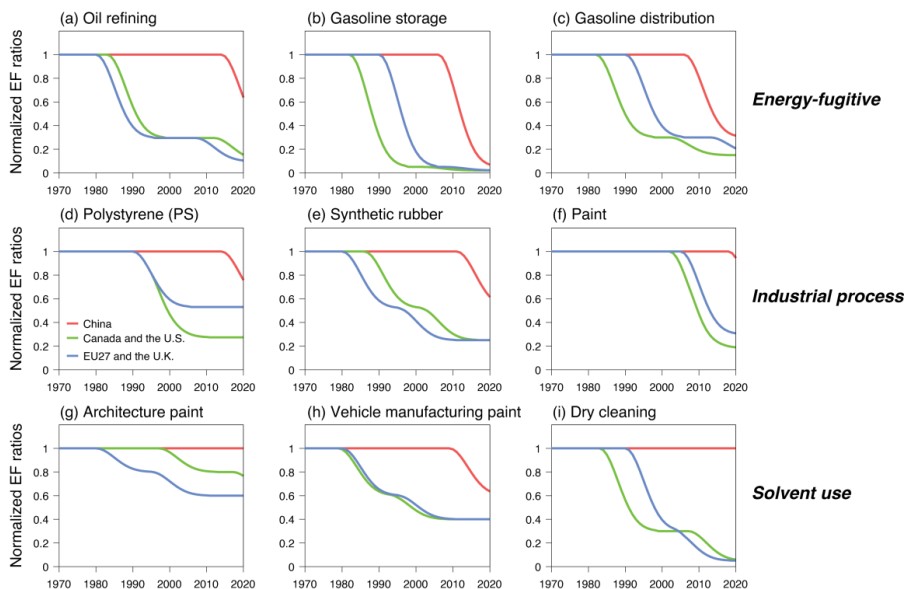

**Figure 3.** Changes in emission factors due to the penetration of control technologies in China (red), Canada and the U.S. (green), and EU27 and the U.K. (blue) for typical sources of energy-fugitive emissions, industrial process, and solvent use. Emission factors are normalized relative to the levels in 1970.





### 2.4.5 Industrial process

Industrial processes were considered for paper/pulp industry, food industry, chemical industry, metal industry, and non-metallic
industry. For paper/pulp and food industries, production statistics were obtained from the Food and Agriculture Organization
(FAO) dataset (FAOSTAT, 2022; See Table S3 for details). FAO data are generally of high quality, providing consistent time
series starting from 1961 and covering all countries, although occasional missing data may exist. These missing data were
filled using linear interpolation for periods of up to 5 years, and moving averages were used for longer gaps to ensure smooth
trends.

For chemical industry, production data were collected from international datasets, including the United Nations (UN) Data,
UN Statistical Yearbooks, and United Nations Framework Convention on Climate Change (UNFCCC) dataset (UNSD, 2022a;
UNSD, 2022b; UNFCCC, 2022). However, these datasets exhibit gaps in time series and country coverage. For example, the
UNFCCC dataset includes production data only for Annex I countries after 1990, and activity rates for certain key sources
(e.g., paint production) are missing for major economies such as China and the U.S. in the UN Data. Consequently, a
reconstruction approach was developed to generate full-coverage and consistent time series for the activity rates of chemical
industry from 1970 to 2020.

To address the gaps, we first supplemented the datasets with additional statistics from literatures, research reports, and official
websites for China, the U.S., EU countries, and other major economies, particularly for large emission sources such as paint
production, the data of which were seriously missing. These supplementary sources are summarized in Table S9. The combined
data from international datasets and supplementary sources were referred to as "actual data" (AD). Next, we reconstructed the
activity rates by mixed effects models. Specifically, when a country had AD for a specific product, the AD were treated as
dependent variables and fitted with GDP and population as independent variables. For countries without AD, the model used
AD, GDP, and population from other countries within the same region and income group (see Table S10 for region and income
group definitions). The fitted models were then applied to generate a full time series of activity rates. Since chemical production
is concentrated in a small number of countries, we used data from the Observatory of Economic Complexity (OEC) to identify
the products that countries can produce (OEC, 2022), thereby avoiding overfitting and reducing biases for countries that
primarily import chemicals rather than produce them. Finally, to constrain the uncertainty at global scale, we collected global
total production statistics if available. From these, we subtracted the total of AD, and then calibrated the total of reconstructed
data to match the remaining global production level. The global data sources are also summarized in Table S9.

A few additional data sources were used for obtaining activity rates of other industrial processes. Ammonia production statistics
in chemical industry, obtained from the United States Geological Survey (USGS) Mineral Yearbooks (USGS, 2022), were
processed in the same way as FAO data. For metal industry, production data for sinter, pellet, and crude steel were obtained
from the Steel Statistical Yearbooks (World Steel Association, 2021). For non-metallic industry, glass production data were
taken from the UN Data, UNFCCC dataset, and official websites of national statistical bureaus in large economies (UNSD,
2022a; UNFCCC, 2022).

Unabated emission factors for industrial process were obtained from the EMEP Guidebook (EEA, 2023), AP-42 database (US
EPA, 1995; 2001), and literatures (Li et al., 2019) as global default values. Country-specific differences in unbated emission
factors were considered for several sources, such as tyre production, as detailed in Supplementary Text S1.

The policy-driven modeling procedures for control technology penetration in industrial process followed the same
methodology as for energy-fugitive emissions in Sect. 2.4.4. Figures 3d-3f illustrate the normalized changes in emission factors
across regions. The trends observed are generally consistent with those for fugitive emissions (Figs. 3a-3c). Policy-driven
emission control had a notable impact on reducing emission factors, although the timing and magnitude of reductions varied
by region and product type.





**2.4.6 Solvent use**

Solvent use includes the consumption of paint, printing ink, glue, and other solvent-consuming activities. The activity rates for paint, printing ink, and glue uses are represented by their apparent consumption, calculated as follows:

$$AC_{i,k,y} = P_{i,k,y} + I_{i,k,y} - E_{i,k,y} \quad (8)$$

where $i$, $k$ and $y$ represent country, product, and year, respectively. $AC$, $P$, $I$, and $E$ represent apparent consumption, production, import, and export, respectively. Production data had been reconstructed as described in Sect. 2.4.5. Trade statistics

from 1988 onward were obtained from the UN Comtrade database (UNSD, 2022c). Due to incomplete temporal coverage in the trade data, apparent consumption by country was first calculated for years when trade data were available. A mixed effects model, as described in Sect. 2.4.5 for chemical industry, was then applied to generate a full time series of apparent consumption for countries with trade data. For countries without trade data, apparent consumption was assumed to be equal to production.

Activity rates for other solvent-consuming activities were mostly represented by proxies due to difficulties in obtaining

statistical consumption data, with the exception of pesticide use. Since emissions from domestic solvent use, dry cleaning, and vehicle resealing were estimated by a population-based method (EEA, 2023), population data from the UN dataset were used as activity rates (UN, 2021). Emissions from vehicle dewaxing were estimated based on vehicle population, the activity rates of which was derived from the fleet-based model in Sect. 2.4.3. Activity rates for degreasing were estimated based on printing ink consumption, following previous studies (Li et al., 2017; Li et al., 2019). Activity rates for pesticide use and wood

preservation were obtained from FAO and UN datasets, respectively.

We used a data-driven turnover approach to differentiate between application purposes and solvent types in paint consumption, representing the substantial gaps in organic content and emission levels across different paint-consuming activities and water-/solvent-borne paints. The activity rates for paint consumption were categorized into six application purposes: architectural interior wall coating, architectural exterior coating, vehicle manufacturing, vehicle repairing, wood coating, and other

industrial use (Li et al., 2019). Regional market segmentation data were collected from literatures and market reports for available years. Missing values were then filled using linear interpolation to obtain a complete time series of category-specific proportions for regional paint consumption from 1970 to 2020. Country-level paint consumption was then allocated into the six categories based on the corresponding regional proportions. Waterborne paint proportions within each application purpose were estimated using the same approach based on regional data and interpolation. Details on regional data are presented in Fig.

S4 and Table S11.

For most solvent-use sources, global default values of unabated emission factors were obtained from the EMEP Guidebook (EEA, 2023), AP-42 database (US EPA, 1995; 2001), and literatures (Li et al., 2019). For paint consumption, purpose- and solvent-type-specific unabated emission factors were applied. For domestic solvent use, dry cleaning, and vehicle resealing, per capita emission factors were used, which varied by country due to differences in socio-economic development and

environmental conditions (EEA, 2023). Specifically, a power-function model was developed for domestic solvent use, using per capita GDP and annual average temperature as independent variables. This model captured the increase in per capita consumption and evaporation of domestic solvents driven by higher income and warmer temperatures. Per capita emission factors for dry cleaning and vehicle resealing were associated with GDP per capita and urbanization rates, respectively (See Supplementary Text S1 for details).

Figure 4 illustrates the evolution of waterborne proportions and the corresponding changes in average unabated emission factors for paint consumption. Global waterborne paint proportions were estimated to increase from approximately 20% in 1970 to nearly 60% in 2020, primarily driven by growth in architectural paint use (Fig. 4a). As a result, the global average unabated emission factor declined by about 33% from 1970 to 2020. All regions experienced decreases in average unabated





emissions factors during this period, although the timing and magnitudes varied (Fig. 4b). In developed regions such as Canada
and the U.S., average emission factors decreased steadily over the entire period, whereas in developing regions like China and
India, the decline became more pronounced after 1990.

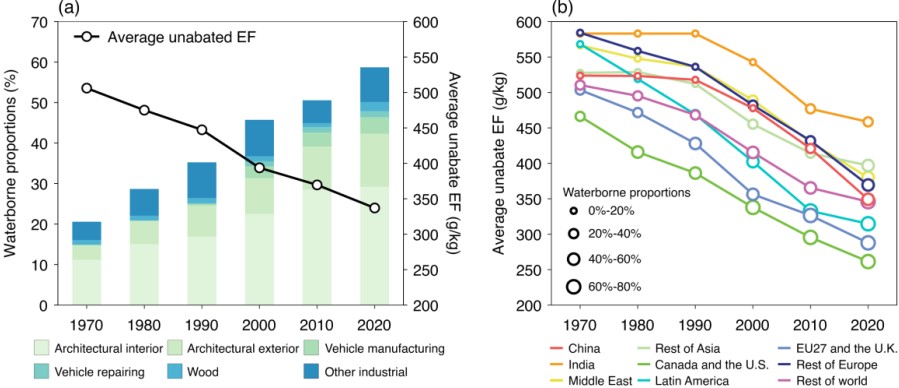

**Figure 4.** (a) Waterborne proportions of global paint use accumulated by application purpose (colored bars) and changes in global average
unbated emission factors (black line with scatters). (b) Changes in regional average unabated emission factors. The sizes of the scatters
represent regional waterborne proportions.

The policy-driven modeling procedures for control technology penetration in solvent use followed the same methodology as
for energy-fugitive emissions in Sect. 2.4.4. Figures 3g-3i show the normalized changes in emission factors for a few solvent-
use sources across regions. For industrial paint uses, such as vehicle manufacturing, emission factors were reduced by up to
60% through technologies like activated carbon absorption and thermal incineration. In contrast, emission control effects were
less pronounced for more scattered sources, such as architectural paint use. From a region-specific perspective, substantial
reductions of emission factors were observed in developed regions., while in developing regions, the effects of control
technologies remained limited for most solvent-use sources.

## 3. Results

### 3.1 Global NMVOC emissions during 1970-2020

Figure 5 presents the evolution of global NMVOC emissions by sector and region. Globally, anthropogenic NMVOC emissions
increased from 71.1 Mt in 1970 to 96.5 Mt in 1990, and remained stable over the next two decades. After 2010, however,
global NMVOC emissions increased again, reaching 112.6 Mt in 2019. In 2020, global emissions declined by 5% due to the
COVID-19 pandemic. Over the 50-year period, contributions from the six major sectors varied (Fig. 5a). The energy-transport
sector (referred to as transport hereinafter) dominated global emission trends for most of the period, though its share gradually
decreased from 39% in 1970 to 28% in 2020. Contributions from the energy-commercial and residential sector and energy-
fugitive emissions (referred to as commercial and residential and fugitive emissions hereinafter) maintained stable, accounting
for 15%-20% and 11%-15% of global emissions, respectively. In contrast, NMVOC emissions from industrial process and
solvent use both increased by approximately 150% during 1970-2020, reaching 12.3 Mt (12%) and 32.5 Mt (31%), respectively,
in 2020. Additionally, contribution from the energy-power and industry sector (referred to as power and industry hereinafter)
remained small (< 5%). Overall, transport, solvent use, and industrial process were the major sectoral drivers of global emission
changes (Fig. 5c).

The dynamics also varied by region (Fig. 5b). In 1970, developed or early-industrialized regions, including Canada and the
U.S., EU27 and the U.K., and rest of Europe, accounted for about 60% of global NMVOC emissions. However, emissions in





these regions declined considerably over the past five decades, particularly from 1990 to 2010 (Fig. 5d), driven by reduced
demand and the implementation of NMVOC emission control policies (see Fig. 2). As a result, these regions contributed only
16% (17.4 Mt) of global emissions in 2020. In contrast, emissions in developing regions grew substantially, fueled by rapid
socio-economic development and limited emission control. For example, NMVOC emissions in India and Latin America
increased by fivefold and threefold, respectively, during 1970-2020. In China, emissions climbed steadily from 1970 to 2010
but plateaued thereafter, with a decreasing trend observed since 2015 due to strengthened clean-air policies (Geng et al., 2021).

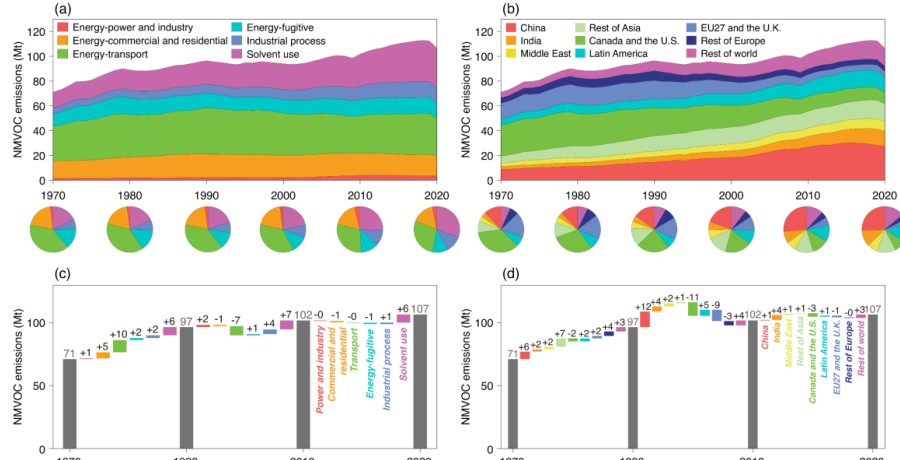


**Figure 5.** (a-b) Global NMVOC emissions from 1970 to 2020, accumulated by (a) sector and (b) region. The pie charts illustrate the sectoral
and regional shares. (c-d) Contributions of sectoral and regional drivers to changes in global NMVOC emissions from 1970 to 2020. The
sector and region mappings are presented in Table S1 and S2, respectively.

Figure 6 shows global NMVOC emissions from different sources within the six major sectors. Coal combustion was the
primary driver of emission changes in the power and industry sector, while biomass combustion dominated emissions in the
commercial and residential sector (Figs. 6a and 6b). In the transport sector, emissions from cars were the major contributor to
the global trend prior to 2010, especially during the decline observed between 1990 and 2010 (Fig. 6c), driven by stricter
emission standards for gasoline-fueled passenger cars worldwide (see Fig. 1). Since 2010, NMVOC emissions from
motorcycles have notably contributed to the global increase, due to the rapid expansion of motorcycle ownership in emerging
economies, such as India and Southeast Asia (Chong et al., 2018; Yan et al., 2024).

For fugitive emissions, oil-related processes were the main drivers of emission changes (Fig. 6d). Venting and flaring
increasingly contributed to emissions (i.e., 39% in 2020), as emission control efforts typically prioritized crude oil refining
and oil product distribution (see Fig. 3), rather than highly fugitive venting and flaring. In the industrial process sector, chemical
and food industries were the dominant contributors to emission increases, together accounting for more than 95% of industrial
process emissions in 2020 (Fig. 6e). The rapid growth of solvent use emissions was fueled by a few sources (Fig. 6f).
Architectural paint use showed the most dramatic increase (i.e., nearly 3% annual average growth rate during 1970-2020) and
became the largest source (8.0 Mt) in 2020, followed by other industrial paint use and domestic solvent use.



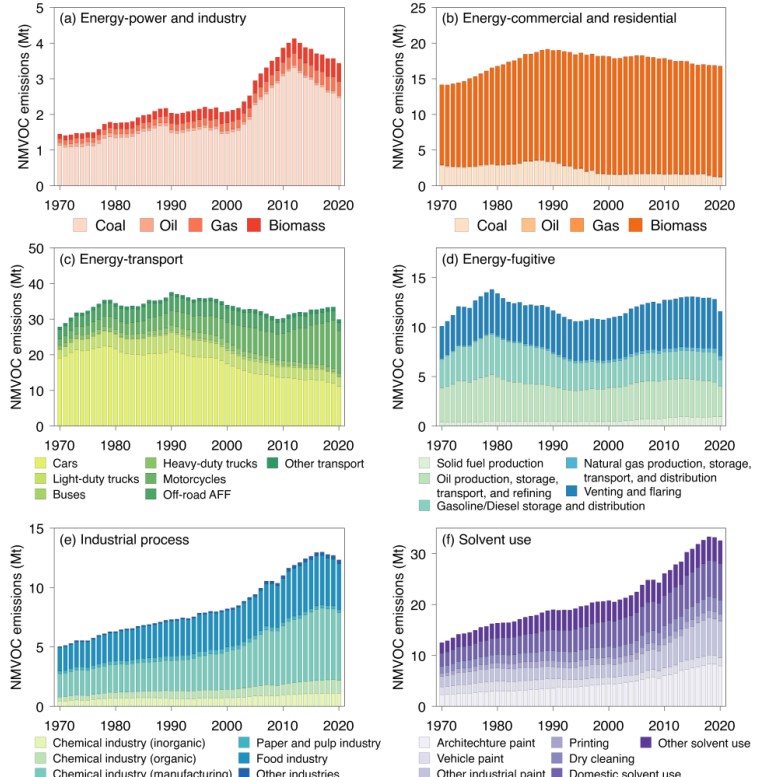

**Figure 6.** Historical NMVOC emissions from 1970 to 2020, accumulated by detailed fuel type or sector for (a) energy-power and industry, (b) energy-commercial and residential, (c) energy-transport, (d) energy-fugitive emissions, (e) industrial process, and (f) solvent use. The sector and region mappings are presented in Table S1 and S2, respectively.

Here, the NMVOC emission structure of chemical industry is further revealed, which encompasses a wide range of products (Table 2). The sources include both basic inorganic and organic chemical industrial processes (IPCC GL category code 2B) and the uses of chemical products in processing and manufacturing (a subset of IPCC GL category code 2D; referred to as manufacturing hereinafter). During 1970-2020, basic inorganic chemicals, basic organic chemicals, and chemical manufacturing accounted for 14%-17%, 12%-17%, and 66%-72% of total emissions from chemical industry, respectively. Emissions from ammonia and carbon black, the two main inorganic products, were comparable. For organic chemicals, however, contributions varied considerably by product. Production of a few types of plastics—PVC, PP, PS, HDPE, and LDPE—accounted for a large portion of NMVOC emissions (i.e., 58% and 76% of emissions from organic chemicals in 1970 and 2020). In chemical manufacturing, emissions from pharmaceutical production quadrupled in 1970-2020, contributing to 25% of total manufacturing emissions in 2020, followed by shoe (19%) and synthetic fiber (12%) manufacturing. The analysis highlights the considerable variability in NMVOC emissions across chemical products, reflecting their distinct emission characteristics and mitigation potential.






**Table 2.** Global NMVOC emissions from chemical industrial process by product in 1970, 1990, 2010, and 2020 (units: kt).

| Sub-sector | Product | 1970 | 1990 | 2010 | 2020 |
|---|---|---|---|---|---|
| Basic inorganic chemicals | Ammonia | 177 | 381 | 514 | 540 |
| | Carbon black | 266 | 281 | 443 | 554 |
| | Total | 443 | 662 | 957 | 1094 |
| Basic organic chemicals | Adipic acid | 5 | 9 | 10 | 9 |
| | Ethylene | 14 | 38 | 42 | 36 |
| | Vinyl chloride | 35 | 35 | 39 | 34 |
| | Styrene | 6 | 13 | 10 | 12 |
| | Low density polyethylene (LDPE) | 30 | 52 | 71 | 83 |
| | High density polyethylene (HDPE) | 23 | 42 | 60 | 73 |
| | Polyvinyl chloride (PVC) | 50 | 134 | 214 | 247 |
| | Polypropylene (PP) | 9 | 43 | 112 | 159 |
| | Polystyrene (PS) | 82 | 175 | 217 | 265 |
| | SAN & ABS resins | 7 | 9 | 8 | 7 |
| | Other synthetic resins | 31 | 43 | 58 | 56 |
| | Ethylene oxide | 12 | 22 | 23 | 23 |
| | Methanol | 28 | 25 | 40 | 76 |
| | Acrylonitrile | 5 | 5 | 3 | 2 |
| | Total | 337 | 645 | 907 | 1082 |
| Manufacturing products | Synthetic rubber | 54 | 66 | 82 | 82 |
| | Tyre production | 63 | 96 | 213 | 220 |
| | Pharmaceutical Production | 339 | 493 | 1203 | 1432 |
| | Asphalt | 202 | 153 | 181 | 252 |
| | Paint production | 190 | 298 | 471 | 657 |
| | Printing ink production | 75 | 145 | 162 | 113 |
| | Glue production | 42 | 96 | 169 | 159 |
| | Shoes production | 394 | 546 | 863 | 1115 |
| | Leather tanning | 1 | 1 | 2 | 2 |
| | Synthetic fibers | 79 | 127 | 486 | 701 |
| | Wool | 7 | 8 | 9 | 9 |
| | Silk | 1 | 1 | 2 | 1 |
| | Cloth | 206 | 316 | 591 | 441 |
| | Artificial fibers | 292 | 199 | 465 | 511 |
| | Total | 1945 | 2545 | 4899 | 5695 |

**3.2 Regional NMVOC emissions**

Figure 7 illustrates the evolution of sector-specific NMVOC emission shares across regions. In developed regions, such as Canada and the U.S. and EU27 and the U.K., the transport sector dominated emissions in the early years. However, its contribution has decreased substantially, following the implementation of improved vehicle standards (see Fig. S2; Yan et al., 2024; ICCT, 2016). In contrast, the shares of emissions from solvent use have nearly doubled (i.e., about 40% in 2020), becoming the largest source of anthropogenic NMVOC emissions in these regions. This trend is consistent with recent

observation-based studies in U.S. cities, which attributed it to surging consumption of volatile chemical products (VCPs) (McDonald et al., 2018; Gkatzelis et al., 2020).





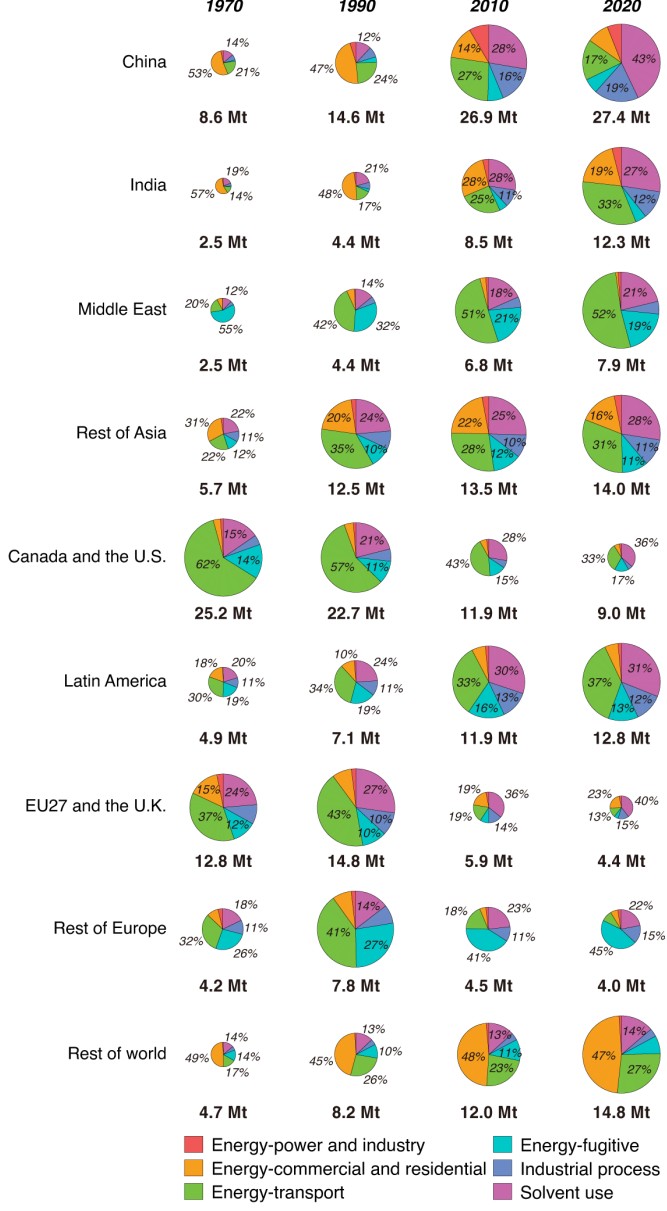

**Figure 7.** Sectoral shares of regional NMVOC emissions in 1970, 1990, 2010, and 2020. The regional emissions are shown below each pie chart. For each region, the relative changes in the radii of the pie charts reflect the proportional changes in emissions.

The patterns in developing regions differed. In China and India, the emission shares from the commercial and residential sector exceeded 50% in 1970, but have sharply declined to less than 20% by 2020. This change was attributed to the increased use of clean fuels (e.g., natural gas) and advanced stoves. In China, emissions from solvent use and industrial process surged, driven by rising demand and delayed mitigation efforts compared to the residential and transport sectors. In India, however, the transport sector experienced rapid emission growth, fueled by an expanding vehicle fleet and weak emission control. Other

regions also exhibited unique characteristics. For example, in the Middle East, the share of fugitive emissions was notably higher than in other regions due to extensive crude oil extraction and production activities. In the rest of world that includes the least developed African countries, the commercial and residential sector maintained dominant (~50%) owing to the





widespread use of biomass in outdated stoves for cooking and heating (Bond et al., 2004).

Here, the NMVOC emission structures are further illustrated across different sources within the six major sectors for four representative regions (Fig. 8). Canada and the U.S. and EU27 and the U.K. represent developed regions; China and India represent upper-middle-income and lower-middle-income developing regions, respectively. In the power and industry sector, the emission shares from biomass and natural gas combustion have steadily increased in the two developed regions (Figs. 8a-8d), reflecting a transition towards low-carbon or clean energy for power generation and industrial production. In the commercial and residential sector, biomass combustion dominated emissions in all regions (Figs. 8e-8h). Notably, EU27 and

the U.K. saw a substantial decline in emission shares from residential coal combustion, with a corresponding increase from residential biomass combustion. This shift was attributed to the electrification of residential end-use consumption in urban areas, which reduced coal use, while biomass remained prevalent in rural areas. This finding is consistent with the increasing attention to residential biomass as a key factor in improving air quality in Europe (van der Gon et al., 2015; Zauli-Sajani et al., 2024). In China, the contribution from coal combustion has declined in recent years due to the replacement of bulk coal with

briquette and natural gas (Zheng et al., 2018; Geng et al., 2021).

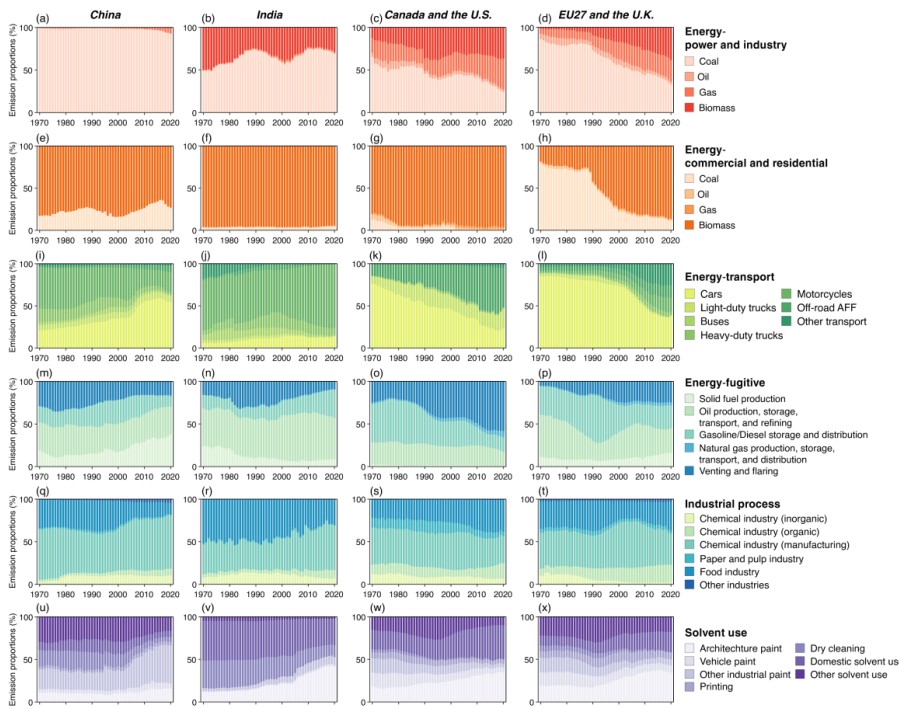

**Figure 8.** The distribution of regional NMVOC emissions by detailed fuel type or sector for (a-d) energy-power and industry, (e-h) energy-commercial and residential, (i-l) energy-transport, (m-p) energy-fugitive emissions, (q-t) industrial process, and (u-x) solvent use in China, India, Canada and the U.S., and EU27 and the U.K. from 1970 to 2020.

Different patterns were observed in the transport sector across regions. In China, emission contributions from cars substantially increased, while contributions from motorcycles declined during 1970-2020 (Fig. 8i). In India, however, emissions from motorcycles dominated throughout the period, steadily increasing since 2000 and accounting for 75% of transport emissions in 2020 (Fig. 8j). This trend resulted from an inadequate road transportation system, heavily reliant on a large motorcycle fleet with minimal emission regulations. In the two developed regions, emission shares from cars have steadily declined, but the

growth in vehicle types differed. In Canada and the U.S., off-road machinery for agriculture, forestry, and fishing saw remarkable increases in emission shares (Fig. 8k). In contrast, in EU27 and the U.K., growth was observed from other transport



sources (e.g., railway, aviation, and navigation) (Fig. 8l). These differences reflect regional variations in the evolution of on-road vehicle fleets and the prevalence of off-road vehicles.

For fugitive emissions, a gap in emission control among the oil-related sources was found in the two developed regions. During the early stages of policy implementation (e.g., NSPS and EU Directives), priority was given to oil refining, storage, and gasoline distribution, while venting and flaring were addressed later (see the timelines in Fig. 2a). As a result, the emission mitigation efforts for these sources were not synchronized. For instance, in the U.S., the current penetration ratios of Stage II abatement technologies (e.g., active vapor recovery systems) for gasoline distribution were estimated to reach nearly 100%, but the adoption of advanced controls for venting and flaring remained limited. Consequently, emission shares from venting and flaring have steadily increased in these regions during 1970-2020 (Figs. 8o and 8p). In contrast, India experienced larger and growing emission shares from oil production, refining, and oil product distribution due to weak control measures (Fig. 8n). In China, emissions from solid fuel production have risen considerably since 2000, driven by the rapid growth in coke production for the iron and steel industry (Fig. 8m).

For industrial process, emission patterns in China and India followed similar trends. Both regions experienced a substantial increase in emission shares from chemical industry, particularly in chemical manufacturing products, while contributions from food industry diminished (Figs. 8q and 8r). In China, however, the growth in chemical industry has leveled off in recent years owing to a series of policies targeting high-emission industries (e.g., GB 27632-2011 for rubber industry; see Fig. 2). In the two developed regions, the emission structures within industrial process remained relatively stable (Figs. 8s and 8t).

In the solvent use sector, emission shares from architectural paint use grew substantially after 2000 in both China and India, driven by the rapid development of the real estate industry (Figs. 8u and 8v). However, a surge in emission shares from other industrial paint use was observed only in China, due to the tremendous expansion of industries such as shipbuilding, railways, and electronics, all of which consumed large amounts of industrial paints. In India, domestic solvent use contributed considerably to emissions (i.e., 30%-47% in 1970-2020), boosted by the large population and high temperatures, which led to increased evaporation rates of VCPs. In the two developed regions, emission patterns were similar, with increased shares from architectural paint use, decreased shares from other paint uses, and notable growth in domestic solvent use (Figs. 8w and 8x). These changes were primarily caused by the asynchrony in control measures for different sources. Control technologies were generally implemented earlier for solvent consumption in well-organized manufacturing activities compared to more dispersed architectural and domestic applications (see the timelines in Fig. 2c), because low-emission technologies, such as electrostatic spraying and activated carbon adsorption, were more technically and economically feasible to implement in industrial production lines and facilities. Consequently, the penetration ratios of control technologies were much higher in industries. The disproportionate distributions of emission control thus reshaped the emission structures within the solvent use sector in developed regions.

### 3.3 Decomposition of driving factors

To further illustrate the effects of different emission parameters (i.e., activity rate, technology distribution, unabated emission factor, and emission control) on global and regional emission changes, we performed a driving factor decomposition analysis. The detailed methodology is provided in Supplementary Text S3. Technology distribution and unabated emission factor were combined as a single driving factor due to their strong interdependence. In the energy-transport sector, unabated emission factors and emission control were integrated in fleet-based turnover models synchronously, which accounted for emission standards. Therefore, the effects of emission mitigation were reflected by changes in technology distributions and emission factors.

Figure 9 presents the driving factors of emission changes from 1970 to 2020 at global scale and in the four representative



regions. Globally, changes in activity rates were the main driving factor of emission increases, largely owing to the production and demand growths in developing regions (Fig. 9a). Conversely, the other two driving factors helped slow the upward trend after 1990, mainly due to reduced fleet-wide emission factors in the transport sector (Fig. 1) and emission abatements in fugitive emissions and solvent use sector (Fig. 3). As a result, the annual average growth rate of global NMVOC emissions was much lower in 1990-2020 (i.e., 0.3%) compared to 1970-1990 (i.e., 1.5%).

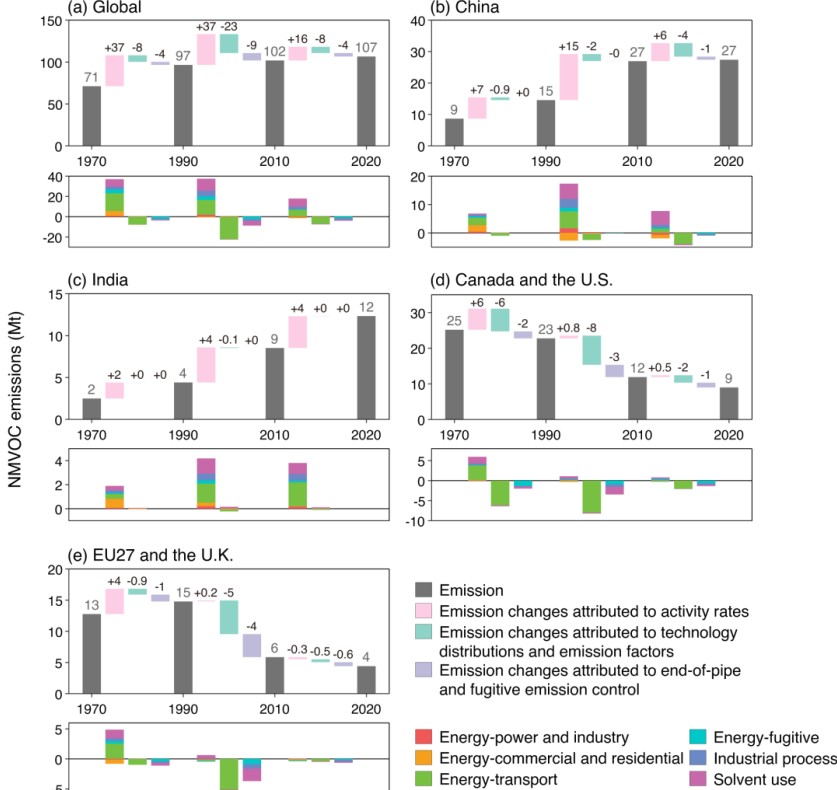

**Figure 9.** Driving factors of emission changes attributed to activity rates (pink bars), technology distributions and unabated emission factors (green bars), and emission control (purple bars) from 1970 to 2020 (a) at global scale, and in (b) China, (c) India, (d) Canada and the U.S., and (e) EU27 and the U.K. Sectoral contributions are shown in the bottom subgraphs.

In developed regions, changes in activity rates were responsible for considerable emission increases during 1970-1990, but had small impact after 1990 due to stable supply and demand (Figs. 9d and 9e). Changes in technology distributions and unabated emission factors led to notable emission reductions, primarily driven by improved emission standards for both on-road and off-road vehicles. The increased adoption of control technologies contributed to emission reductions in fugitive emissions and solvent use sector, especially during 1990-2010. This was promoted by a series of standards (e.g., NESHAP) and directives implemented during this period to mitigate ozone and HAPs pollution (see Fig. 2).

In developing regions, changes in activity rates substantially drove emission increases across almost all sectors (Figs. 9b and 9c). However, in China, changes in activity rates in the commercial and residential sector contributed to emission reductions since 1990 as a result of decreased consumption of emission-intensive biomass. Changes in technology distributions and unabated emission factors led to large emission decreases in China (e.g., 4.3 Mt in 2010-2020), driven by improved vehicle standards (see Fig. S2) and increased market share of waterborne paints. The shift in paint types was facilitated by standards such as GB 18582-2008 for architectural interior wall coating and GB 24409-2009 for automobile manufacturing. Emission



control measures in China helped reduce emissions after 2010, although their effects were not very pronounced, as most
policies were implemented after 2015 and the penetration ratios of control technologies remained low or moderate in most
industries by 2020 (see Fig. 3). During 2010-2020, the emission reduction driven by changes in technology distributions,
unabated emission factors, and emission control nearly offset the emission increases driven by activity rates. Conversely, in
India, where national efforts to mitigate NMVOC emissions have not yet been initiated, changes in the driving factors other
than activity rates had negligible effects on emission changes.

## 4. Discussions

### 565    4.1 Uncertainties and limitations

The NMVOC emission estimates are subject to uncertainties arising from several factors, including lack of reliable activity
rates for scattered areal sources, emission factors that vary with procedures and practices, and complex control measures (Wu
et al., 2016; Hoesly et al., 2018). In this study, emission uncertainties were quantified by a Monte Carlo approach, following
previous studies (Zhao et al., 2011; Chen et al., 2022). The term "uncertainty" here refers to the 95% confidential interval (CI)
around the central estimates. The emission parameters and their statistical distributions were placed in a Monte Carlo
framework, and 1,000 simulations were performed to quantify the emission uncertainties. The probability distributions of
emission parameters and other details of the uncertainty analysis are presented in Table S12 and Supplementary Text S4.

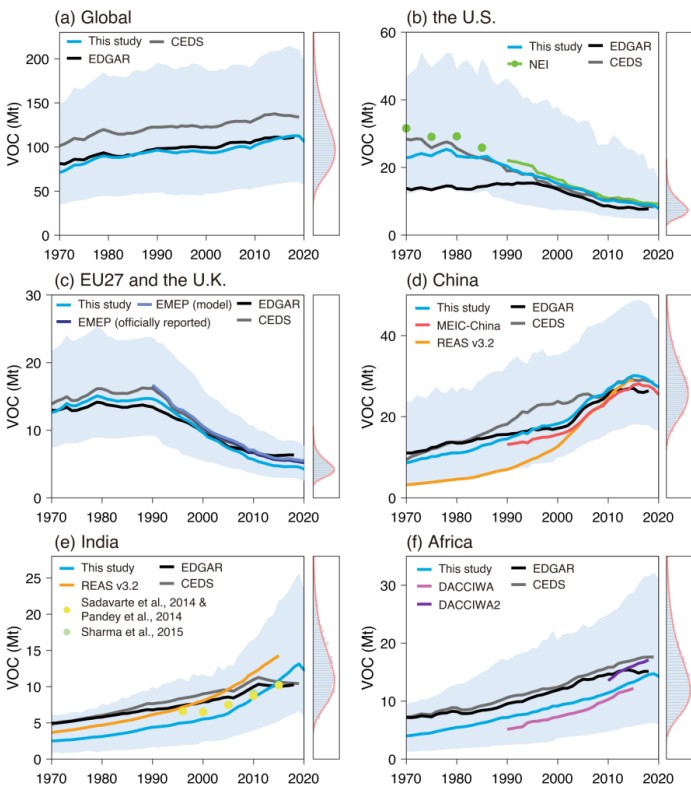

**Figure 10.** Comparison of NMVOC emissions between this study and other inventories (a) at global scale and in (b) the U.S., (c) EU27&
the U.K., (d) China, (e) India, and (f) Africa. The light blue shaded areas represent the uncertainty ranges of emission estimates. The right
subgraphs show the probability distributions of emission estimates in 2020. Note that the minor sources that are not included in this study
(e.g., waste treatment) are excluded in other inventories for comparison within the same boundary. EDGAR version 6.1 and CEDS version
2021-04-21 are used here. MEIC-China only provides data for Chinese mainland.



Globally, the uncertainty ranged from -52% to +109% in 1970 and from -47% to +85% in 2020 (Fig. 10a). The uncertainty
range narrowed over time due to improvements in statistical reporting systems for activity rates and more complete information
on other parameters. In developed regions, uncertainties were generally better constrained because of higher-quality data. For
example, in EU27 and the U.K., the uncertainty ranged from -(45%–35%) to +(60%–80%) during most of the 1970-2020
period (Fig. 10c). In developing regions, uncertainties were typically larger. In China and India, uncertainties varied from -
(64%–38%) to +(60%–178%) and -(68%–49%) to +(89%–145%), respectively, with narrowed ranges over time (Figs. 10d
and 10e). In Africa, a broader uncertainty range was observed because of the dominant contribution from residential fuel
combustion, which was highly uncertain (Fig. 10f). Compared to previous studies, the uncertainties here were comparable in
most regions but lower in certain regions like India (Zhang et al., 2009; Kurokawa et al., 2013; Crippa et al., 2018). This could
be attributed to different methods used for uncertainty analysis (e.g., a standard-deviation-based approach was used in EDGAR
emission inventory; Crippa et al., 2018) as well as varying assumptions regarding probability distributions.

This study also bears a few limitations, which could be addressed in future work. First, the current emissions exclude open
biomass burning, which may be a non-negligible source in some developing countries (Pandey and Sahu, 2014; Yin et al.,
2019). A few minor sources, such as waste treatment, are also not included (Sharma et al., 2019). Second, since the models
developed for technology distributions and emission control are mostly region-specific, future efforts could focus on better
reflecting country-specific variations. Third, NMVOC emissions from industry and solvent use are assumed to be uncontrolled
in developing countries, except China, which may not be consistent with the actual situation in some countries. Therefore, the
policy-driven turnover model could be extended spatially based on a broader investigation in the future.

**4.2 Comparison with other inventories**

We compared the NMVOC emissions in this study (referred to as MEIC-global-NMVOC) with several widely-used inventories,
including the Emission Database for Global Atmospheric Research (EDGAR) and the Community Emission Data System
(CEDS) at global and regional scales (Crippa et al., 2018; Hoesly et al., 2018), the National Emission Inventory (NEI) for the
U.S. (EPA, 2023), the European Monitoring and Evaluation Programme (EMEP) inventory for EU27 and the U.K.
(EMEP/CEIP, 2023), MEIC-China and the Regional Emission inventory in Asia version 3.2 (REAS v3.2) for China (Li et al.,
2019; Kurokawa and Ohara, 2020), REAS v3.2 and two local studies for India (Sadavarte et al., 2014; Pandey et al., 2014;
Sharma et al., 2015), and the Dynamics-Aerosol-Chemistry-Cloud Interactions in West Africa (DACCIWA) inventory for
Africa (Keita et al., 2021).

Globally, the overall emission patterns are consistent across the inventories, although EDGAR and CEDS show slight emission
increases during 1990-2010, while MEIC-global-NMVOC show a more stable trend, with a more rapid increase observed after
2010 (Fig. 10a). Specifically, MEIC-global-NMVOC estimates are generally close to those of EDGAR (i.e., relative
differences range from -12.6% to +1.7%) and approximately 16%-30% lower than those of CEDS. The differences are mainly
found in sectors such as transport, fugitive emissions, and solvent use (Figs. 11a and S5a), which are likely attributable to
inconsistencies in source categories, treatment of activity rates, selection of emission factors, and representation of
technologies (Li et al., 2017). Additionally, it should be noted that EDGAR and CEDS may include NMVOC emissions from
coal mining (a subset of IPCC GL category code 1B1), which are not considered in this study, potentially leading to large
differences in fugitive emissions. This source, characterized by a high degree of uncertainty, may have a substantial impact on
total emissions (EEA, 2023), but it cannot be isolated from the available data in EDGAR and CEDS. To address this, we also
compared global emissions excluding all fugitive emissions from solid fuel production (Fig. S6). It was found that the emission
gap between MEIC-global-NMVOC and CEDS narrowed, especially in recent years, although the trends after 2010 became
more inconsistent.

For the U.S, MEIC-global-NMVOC estimates are lower than NEI (i.e., relative differences ranging from -27.6% to -1.9%),



showing consistent decreasing trends and narrowing gaps after 1990 (Fig. 10b). The discrepancies in the early years are mainly attributed to differences in the transport and solvent use sectors (Figs. 12a and S7a). After 2000, the differences are mainly caused by a sharp increase in fugitive emissions in NEI, which is difficult to capture in this study, likely due to methodological inconsistencies in NEI. CEDS also shows consistent trends for the U.S., but EDGAR estimates are much lower, especially during the 1970-1990 period, mainly due to discrepancies in the transport and solvent use sectors (Figs. 11b and S5b). For

EU27 and the U.K., MEIC-global-NMVOC estimates are generally close to EMEP inventory (i.e., relative differences ranging from -22.2% to -6.3%), as well as CEDS and EDGAR, although slightly lower estimates are observed in recent years (Fig. 10c). The differences are primarily found in the solvent use, commercial and residential, and transport sectors (Figs. 11c, 12b, S5c, and S7b).

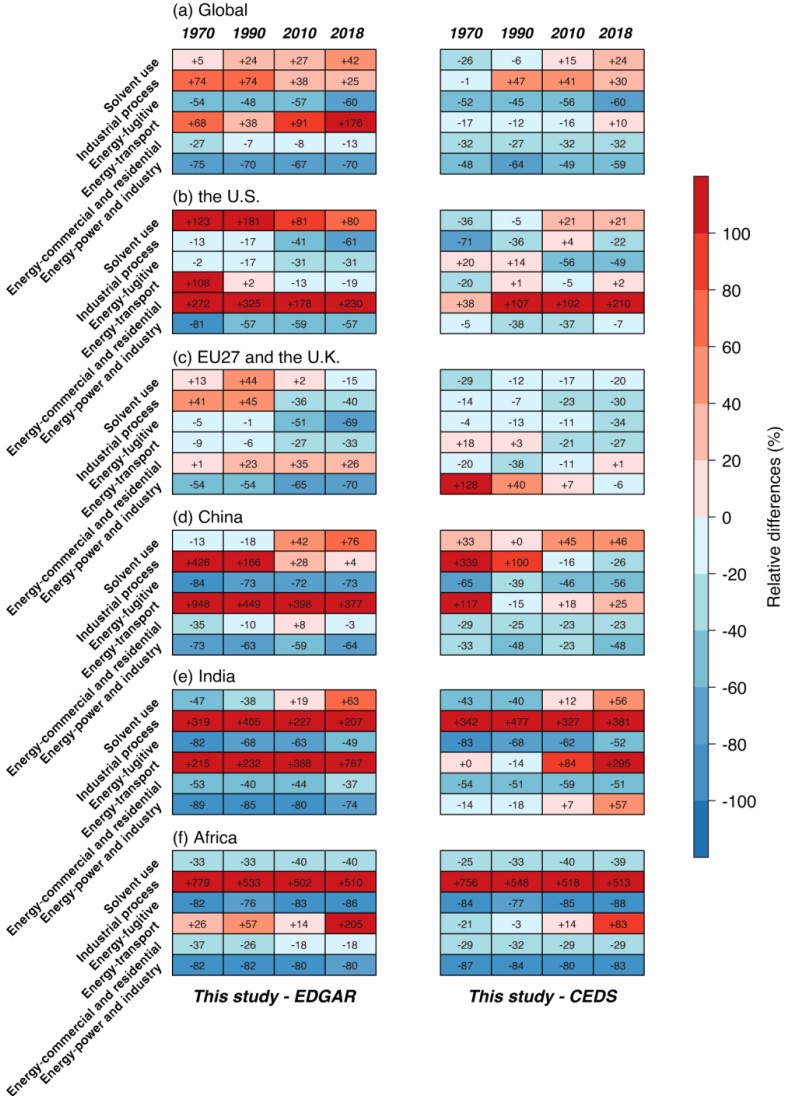

**Figure 11.** Relative differences in sectoral NMVOC emissions (a) at global scale and in (b) the U.S., (c) EU27 and the U.K., (d) China, (e) India, and (f) Africa between EDGAR (left column) or CEDS (right column) inventories and this study. Note that off-road vehicles for agriculture, forestry, and fishing are grouped under the commercial and residential sector for comparison here, in alignment with the definitions in EDGAR and CEDS (i.e., IPCC GL category 1A4).



For China, MEIC-global-NMVOC emissions are in close agreement with MEIC-China (Fig. 10d), as the data sources of
activity rates, selection of emission factors, and representation of technologies are similar. In the early years, EDGAR and
CEDS report higher emissions, while REAS estimates are much lower. However, the gaps among the inventories narrowed
substantially after 2010. A more detailed comparison between MEIC-China and other inventories can be found in our previous
studies (Li et al., 2017; 2019). In India, MEIC-global-NMVOC estimates are 24%-35% lower than REAS, and align more
closely with two local studies, especially after 2010 (Fig. 10e). EDGAR and CEDS show slower emission growth compared
to MEIC-global-NMVOC and REAS. The activity rates of residential biomass combustion, including both the total amount
and the fuel type structure, are likely a major source of emission discrepancies in India (Paliwal et al., 2016; Kurokawa and
Ohara, 2020; Stewart et al., 2021). Additionally, the divergent trends between this study and CEDS or EDGAR in the recent
decade result from varying growth rates in the transport sector (Figs. 11e and S5e). For Africa, emissions in this study are
higher than DACCIWA but show a narrowing difference in recent years (e.g., +8.8% in 2015), while DACCIWA2, EDGAR,
and CEDS report higher emissions (Fig. 10f). Fugitive emissions and the commercial and residential sector contribute most to
these differences (Figs. 11f and S5f).

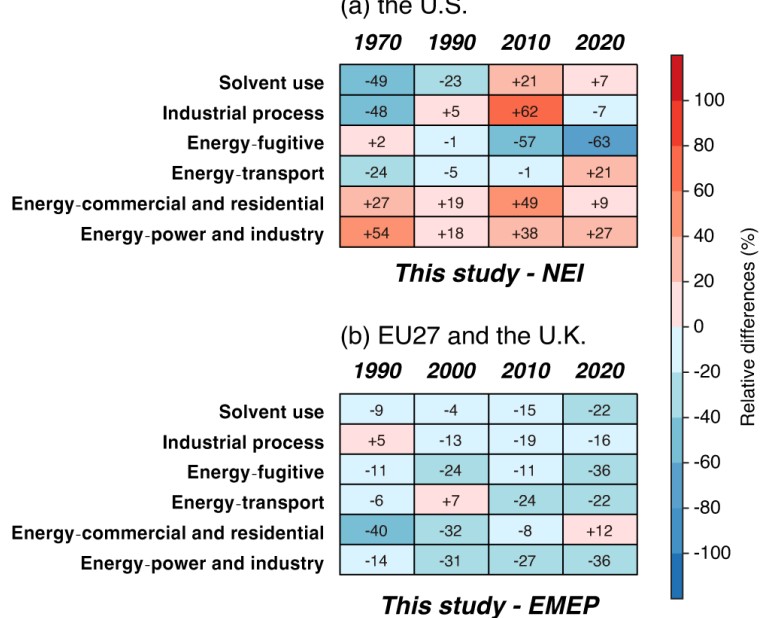

**Figure 12.** Relative differences in sectoral NMVOC emissions in (a) the U.S. and (b) EU27 and the U.K. between regional inventories (i.e.,
NEI and EMEP) and this study.

## 5. Concluding remarks

In this study, we developed a new technology-based global NMVOC emission inventory under the bottom-up framework of
MEIC, to better represent the evolution and drivers of NMVOC-emission-related technologies and global NMVOC emissions.
To address the gaps in activity rates, particularly for chemical industry and solvent use, we developed data fusion and
reconstruction approaches to generate consistent, full-coverage time series of activity rates. Next, we modeled the evolution
of technology distributions and emission control measures using both data-driven and policy-driven technology turnover
models. Global anthropogenic NMVOC emissions from 1970 to 2020 were then estimated incorporating emission factors
derived from international databases, local literatures, and emission models.



Based on the developed inventory, we tracked the dynamics of global NMVOC emissions across sectors, fuel types, and specific products, revealing distinct and evolving contributions from various emission sources. The emission structures in key regions were analyzed, highlighting region-specific dynamics that were closely tied to local socio-economic development, technology advancements, and environmental policies. For instance, in developing regions, NMVOC emission growths were driven by surges in activity rates, but curbed by adoption of advanced combustion technologies, improved vehicle standards, and substitution of solvent-borne paints. In contrast, in developed regions, in addition to these mitigation measures, emission control technologies substantially reduced fugitive emissions from energy production and emissions from solvent use, driven by the implementation of policies such as the U.S. NSPS, NESHAP, and EU Directives. Despite large uncertainties, emission estimates in MEIC-global-NMVOC were generally comparable with other emission inventories at aggregated level for global or regional total, but notable differences were observed at sectoral level. This underscores the need for future efforts to better clarify the key influencing factors behind emission differences and uncertainties.

In the future, additional data products, including speciated emissions, chemical-mechanism-based emissions, and high-resolution emission grids, will be developed to support atmospheric chemical transport models. We expect that the inventory will serve as a high-quality, open-access data foundation for air pollution research and mitigation, helping to set up cost-effective measures and reduce human exposures to $PM_{2.5}$, $O_3$, and HAPs.

**Conflict of interest**

The authors declare that they have competing interests as follows: At least one of the co-authors is a member of the editorial board of Atmospheric Chemistry and Physics.

**Acknowledgements**

This work is supported by the National Key R&D program of China (grant no. 2022YFC3700605). We thank Dr. Meng Li for helpful discussion and cooperative development of MEIC-global-NMVOC.

**Author contributions**

Q.Z. designed the study. R.X. performed data set construction and emission estimates with support from H.M. and J.L. on emission estimates of fugitive emissions, industrial process, and solvent use, from L.Y. and L.W. on emission estimates of transport, from X.Q., X.Y., and Y.F. on emission estimates of power and industry, from H.H., N.W., Y.W., and H.W. on data compilation, from X.L. and Q.X. on calculation system, and from D.T., G.G., and K.H. on data analysis. R.X. and Q.Z. interpreted the data. R.X. and Q.Z. wrote the paper with input from all co-authors.

**Data availability**

The MEIC-global-NMVOC emission data used in this study are publicly available at https://zenodo.org/records/14978125.

**Code availability**

The code used to manipulate the data and generate the results is available from the corresponding author upon reasonable request.



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
