# Peer review of "A technology-based global non-methane volatile organic compounds (NMVOC) emission inventory under the MEIC framework"

_EGUsphere, 2025_

## Author Comment (AC1)

**Reviewer(s)' Comments to Author:**

**Reviewer: 1**

**General comments**

This work presents a very detailed and thorough technology-based global NMVOC emission evolution inventory 1970-2020, which is also compared to other widely used inventories. Overall, the quality of the presented work is very high.

**Response**: We thank the Referee for the positive view on the manuscript and the thoughtful suggestions below. We've made a number of revisions in response, and believe the manuscript has been improved. A point to point response is presented below.

**Specific comments**

Introduction: The authors should add at least a paragraph discussing in some detail the representation of the speciation of NMVOC and differences in lumping in different inventories, including the presented one.

**Response**: We thank the Referee for the thoughtful comment. As suggested by the Referee, we have added a paragraph in the Introduction to describe NMVOC speciation and lumping for different chemical mechanisms, as well as their relationship with total NMVOC emission estimates. Relevant references are also cited.

*"NMVOC emission estimates also provide the basis for emission data used in chemical transport models (CTMs) by further developing chemically-resolved NMVOC speciation and subsequent lumped emissions with different chemical mechanisms (Li et al., 2019; Huang et al., 2017). Therefore, the biases and uncertainties in total NMVOC emission estimates can be compounded by those in speciation profiles and mechanism-specific mapping tables (Huang et al., 2017; von Schneidemesser et al. 2023), affecting accurate emission estimates of key species such as HAPs (Pye et al., 2023; Li et al., 2019) and modeling performance of $O_3$ and SOA in CTMs (von Schneidemesser et al., 2016; Rowlinson et al., 2024; Oliveira et al., 2025)."*

However, it should be noted that this study focuses on the estimates and evaluation of total NMVOC emissions and does not include speciation or lumping methodology/analyses. This allows us to present a detailed description of the methodology and a comprehensive analysis for total NMVOC emission estimates without making the manuscript overly lengthy and complex.

We are currently working on NMVOC speciation and lumping to chemical mechanisms in the MEIC-global-NMVOC emission inventory, with preliminary results prepared for a separate manuscript. We first collect speciation profiles by integrating existing profile datasets and new measurements (e.g., Simon et al., 2010; Theloke and Friedrich, 2007; Li et al., 2014). These profiles are then mapped to each emission source, and uncertainties are reduced by developing

composite profiles with corrections for OVOC fractions (Li et al., 2014). NMVOC emissions are speciated to derive chemically-resolved emissions using the source-specific emission estimates presented in this work and the developed profiles. Finally, we apply the mapping tables developed by Carter (2023) to lump the chemically-resolved emissions into chemical-mechanism-specific emissions that can be used directly by different chemical transport models.

We are committed to presenting the detailed methodology and results of NMVOC speciation and lumping in a separate manuscript. We hope the Referee understands our considerations for addressing these efforts separately. We also acknowledge that the lack of speciation and lumping analysis is a limitation of the current study and have added a brief discussion in Section 4.1 of the revised manuscript as below.

*"Finally, NMVOC emission speciation is not included in this study, but will be addressed in future work to provide species-resolved emissions and model-ready emissions for different chemical mechanisms."*

References:

Simon, H., Beck, L., Bhave, P. V., Divita, F., Hsu, Y., Luecken, D., Mobley, J. D., Pouliot, G., Reff, A., Sarwar, G., and Strum, M.: The development and uses of EPA's Speciate database, Atmos. Pollut. Res., 1(4), 196-206, 2010.

Theloke, J., and Friedrich, R.: Compilation of a database on the composition of anthropogenic VOC emissions for atmospheric modeling in Europe, Atmos. Environ., 41(19): 4148-4160, 2007.

Li, M., Zhang, Q., Streets, D. G., He, K. B., Cheng, Y. F., Emmons, L. K., Huo, H., Kang, S. C., Lu, Z., Shao, M., Su, H., Yu, X., and Zhang, Y.: Mapping Asian anthropogenic emissions of non-methane volatile organic compounds to multiple chemical mechanisms, Atmos. Chem. Phys., 14, 5617–5638, 2014.

Carter, W. P. L.: Development of an improved chemical speciation database for processing emissions of volatile organic compounds for air quality models, https://intra.engr.ucr.edu/~carter/emitdb/, 2023.

Section 4.2 comparison with other inventories: Do all the compared inventories use the same speciation/lumping for NMVOC? If not, the authors should discuss also what this might imply for the comparison.

**Response**: We thank the Referee for the thoughtful comment. As mentioned above, this study primarily focuses on the estimates and analysis of total NMVOC emissions. Therefore, speciation and lumping processes do not influence the comparative results of total NMVOC emissions presented in this manuscript. Nevertheless, as the Referee points out, the differences in total NMVOC emissions can be further compounded by the differences in speciation profiles and chemical-mechanism-specific mapping tables in the subsequent steps of NMVOC speciation and lumping. To reflect this important point, we have added a paragraph in Section

4.2 discussing the differences in total NMVOC emissions, speciation, and lumping, as well as their implications.

*"Importantly, the discrepancies in NMVOC emission estimates among the emission inventories can propagate through subsequent steps of chemical speciation and chemical-mechanism-specific lumping, ultimately affecting emission estimates of key chemical species and the performance of CTMs in simulating secondary air pollutants (Pye et al., 2023; von Schneidemesser et al., 2016). Therefore, it is crucial for developers of NMVOC emission inventories to collaboratively identify and diagnose the underlying causes of these discrepancies and to better constrain source-specific total NMVOC emission estimates before the differences are further amplified by the application of different speciation profiles and mapping tables for chemical-mechanism-specific lumping (Huang et al., 2017; von Schneidemesser et al. 2023)."*

As mentioned above, the detailed comparison and evaluation related to speciation and lumping will be presented in a separate manuscript. We hope the Referee understands our considerations for addressing these efforts separately.

Line 235, "as higher temperatures lead to more NMVOC evaporation": here the authors might e.g. cite Kourtidis K.A., I.C. Ziomas, B. Rappenglueck, A. Proyou, D. Balis, Evaporative traffic hydrocarbon emissions, traffic CO and speciated HC traffic emissions from the city of Athens, Atmospheric Environment 33(23), 3831-3842, 1999, https://doi.org/10.1016/S1352-2310(98)00395-1

**Response**: We thank the Referee for the constructive suggestion. The references have been supplemented as shown below.

*"For start and evaporation modes, the effects of ambient temperature were considered in the calculation, as higher temperatures lead to more NMVOC evaporation (Kourtidis et al., 1999; Rubin et al., 2006), ..."*

References:

Kourtidis, K. A., Ziomas, I. C., Rappenglueck, B., Proyou, A., and Balis, D.: Evaporative traffic hydrocarbon emissions, traffic CO and speciated HC traffic emissions from the city of Athens, Atmos. Environ., 33(23), 3831-3842, 1999.

Rubin, J. I., Kean, A. J., Harley, R. A., Millet, D. B., and Goldstein, A. H.: Temperature dependence of volatile organic compound evaporative emissions from motor vehicles, J. Geophys. Res. Atmos., 111(D3), 2006.

Fig. 2 in page 9: This appears more like a Table than a Figure.

**Response**: We thank the Referee for the constructive comment. As the Referee suggested, Figure 1 could be presented as a table. However, we believe that using different colors and shading in the figure helps to distinguish regions and policy stages, thereby improving the

readers' understanding of policy timelines. Since ACP tables are required to be black and white, we have retained the figure format.

**Technical corrections**

Line 26: Change "growths were" to "growth was".

Line 55: We feel that the authors by "are generally not presented in detail" mean "are generally not detailed enough"

Line 58: Change "should be concerned" to "should be addressed" and "First" to "Firstly".

Line 60: Change "chemical" to "the chemical".

Line 61: Change "Second" to "Secondly".

Line 76, 158, 279, and elsewhere in the text: It is not clear what the authors mean by the plural "literatures".

Line 141and footnote of Table 1: Change "synchronously" to "simultaneously".

Lines 158-159: "market surveys for available years" means "market surveys for the years such surveys were available"?.

Line 192: Change "stove type proportion" to "stove type percentages".

Line 194: Change "the besting performing" to "the best performing".

Line 209: Change "penetration of advanced stove" to "penetration of advanced stoves".

Line 255: Change "scatters" to "circles".

In Fig. 3, a legend with the different line colors would help.

Line 401: Change "maintained stable" to "remained stable".

**Response**: We thank the Referee for the helpful and detailed suggestions. The technical errors have all been corrected in the revised manuscript.

---

## Author Comment (AC2)

**Reviewer(s)' Comments to Author:**

**Reviewer: 2**

The manuscript by Xu et al. presents a new global NMVOC emission inventory, the MEIC-global-NMVOC database, which offers a valuable data source for atmospheric chemistry modeling and policymaking. This work is important as it provides a very detailed representation of NMVOC-emission-related technologies and their impacts on emission changes, addressing a gap in most previous global-scale studies. The authors also conducted comprehensive comparisons across multiple emission inventories, elucidating key differences. The manuscript is well-written, with high-quality figures and thorough analysis. I recommend it for publication in *Atmospheric Chemistry and Physics*, pending minor revisions.

**Response**: We thank the Referee for the positive and accurate summary of our work and for the fair and thoughtful comments below. We've made a number of revisions in response, and believe the manuscript has been improved. A point to point response is presented below.

**Introduction**

1. I suggest that the authors cite several relevant previous studies on NMVOC emission inventories to provide better context for their work. For example:

Bo, Y., Cai, H., & Xie, S. D. (2008). Spatial and temporal variation of historical anthropogenic NMVOCs emission inventories in China. *Atmospheric Chemistry and Physics*, *8*(23), 7297-7316.

Sharma, S., Goel, A., Gupta, D., Kumar, A., Mishra, A., Kundu, S., ... & Klimont, Z. (2015). Emission inventory of non-methane volatile organic compounds from anthropogenic sources in India. *Atmospheric Environment*, *102*, 209-219.

Zhao, Y., Mao, P., Zhou, Y., Yang, Y., Zhang, J., Wang, S., ... & Li, W. (2017). Improved provincial emission inventory and speciation profiles of anthropogenic non-methane volatile organic compounds: a case study for Jiangsu, China. *Atmospheric Chemistry and Physics*, *17*(12), 7733-7756.

Li, M., Kurokawa, J., Zhang, Q., Woo, J. H., Morikawa, T., Chatani, S., ... & McDonald, B. C. (2024). MIXv2: a long-term mosaic emission inventory for Asia (2010–2017). *Atmospheric Chemistry and Physics*, *24*(7), 3925-3952.

**Response**: We thank the Referee for the constructive suggestion. The studies suggested by the Referee have been added in the manuscript as shown below.

*"...and larger gaps can be found for major emitters and specific sectors (Bo et al., 2008; Li et al., 2019; Sharma et al., 2015; Zhao et al., 2017; Kurokawa and Ohara, 2020; Li et al., 2024), highlighting the large uncertainties in current NMVOC emission estimates."*

2. Line 38: When discussing the impacts of NMVOC emissions on air quality and health, it would be better to cite and discuss some recent studies. For example:

Xiong, Y., Du, K., & Huang, Y. (2024). One-third of global population at cancer risk due to elevated volatile organic compounds levels. *npj Climate and Atmospheric Science*, *7*(1), 54.

Qin, M., She, Y., Wang, M., Wang, H., Chang, Y., Tan, Z., ... & Hu, J. (2025). Increased urban ozone in heatwaves due to temperature-induced emissions of anthropogenic volatile organic compounds. *Nature Geoscience*, 1-7.

Partha, D. B., Xiong, Y., Prime, N., Smith, S. J., & Huang, Y. (2025). Long-term impacts of global solid biofuel emissions on ambient air quality and human health for 2000–2019. *GeoHealth*, *9*(3), e2024GH001130.

**Response**: We thank the Referee for the constructive suggestion. The recent studies suggested by the Referee have been cited and discussed in the manuscript as shown below.

*"Thus, NMVOC have substantial global impacts on air quality, climate, and human health (Fry et al., 2014; Laurent and Hauschild, 2014; Li et al., 2019; Xiong et al., 2024; Partha et al., 2025). Previous studies have demonstrated that mitigating anthropogenic NMVOC emissions is the key to controlling $O_3$ and fine particulate matter ($PM_{2.5}$) pollution, as well as addressing other negative effects in various regions (West et al., 2007; Fry et al., 2014; Sharma et al., 2016; Hendriks et al., 2016; Li et al., 2019; Qin et al., 2025)."*

3. Line 57: "also contribute to divergent emission estimates" would be better.

**Response**: We thank the Referee for the constructive suggestion. The sentence has been revised as suggested.

4. Line 66: The authors mentioned "their validation with observations", but this issue is not addressed in their work. I recommend the authors to revise this statement to avoid potential confusion.

**Response**: We thank the Referee for the thoughtful comment. We have revised the sentence as below to avoid potential confusion.

*"Third, comparisons among global NMVOC emission inventories are limited, which hinders understanding of their discrepancies and underlying causes."*

**Methods and data**

1. Line 110: I think the square brackets in equation (1) should be parentheses?

**Response**: We thank the Referee for the detailed suggestion. We have revised the equation as below.

$$E_i = \sum_j \sum_k \left[ A_{i,j,k} \times \left( \sum_m X_{i,j,k,m} \times EF_{i,j,k,m} \right) \right] \quad (1)$$

2. Line 178: In section 2.2, 228 countries are defined. Why are there only 208 countries here?

**Response**: We thank the Referee for the thoughtful question. The activity rates for energy consumption are available for 208 countries, consistent with the country coverage of energy-related data sources such as the International Energy Agency (IEA), as illustrated in our previous work on MEIC-global-$CO_2$ emission inventory (Xu et al., 2024). However, the population data used as the activity rates for certain non-energy-related sources (e.g., domestic solvent use) include an additional 20 small countries/territories. Therefore, the total number of countries/territories included in this study is 228. We have clarified this point in Section 2.2 as below.

*"MEIC-global-NMVOC includes 228 countries/territories, an extension from the 208 countries/territories in the MEIC-global-$CO_2$ inventory (Xu et al., 2024). The additional 20 countries are included because the population data used as the activity rates of a few solvent-consuming sources cover more small countries than the energy statistics used in MEIC-global-$CO_2$."*

References:

Xu, R., Tong, D., Xiao, Q., Qin, X., Chen, C., Yan, L., Cheng, J., Cui, C., Hu, H., Liu, W., Yan, X., Wang, H., Liu, X., Geng, G., Lei, Y., Guan, D., He, K. and Zhang, Q.: MEIC-global-$CO_2$: A new global $CO_2$ emission inventory with highly-resolved source category and sub-country information, Sci. China Earth Sci., 67, 450-465, 2024.

3. Line 190: I am curious about what "primary solid biofuel" includes. Are wood, crop residue, or other types of solid fuels distinguished here? If not, please clarify it.

**Response**: We thank the Referee for the thoughtful question. Primary solid biofuels refer to any plant- or waste-based materials used directly as fuel, including firewood, crop residues, animal waste, sawdust, and other solid biofuels. Due to limitations in data availability, this study does not distinguish between different types of primary solid biofuels; instead, they are used collectively as a single category. We have clarified it in Section 2.4.2 as follows.

*"It should be noted that different types of primary solid biofuels—such as wood, crop residues, and animal waste—were not distinguished in this study."*

4. Line 234-237: Some references should be supplemented about the impacts of different factors on vehicle emission factors.

**Response**: We thank the Referee for the constructive suggestion. The references have been supplemented as shown below.

*"For start and evaporation modes, the effects of ambient temperature were considered in the calculation, as higher temperatures lead to more NMVOC evaporation (Kourtidis et al., 1999; Rubin et al., 2006), while lower temperatures increase start-mode emission factors due to longer time required for the catalyst to reach operating conditions (Singer et al., 1999; Reiter and Kockelman, 2016). For running emissions, vehicle deterioration was considered, which results in increased emission factors as cumulative mileage rises (Chiang et al., 2008; Borken-Kleefeld and Chen, 2015)."*

References:

Kourtidis, K. A., Ziomas, I. C., Rappenglueck, B., Proyou, A., and Balis, D.: Evaporative traffic hydrocarbon emissions, traffic CO and speciated HC traffic emissions from the city of Athens, Atmos. Environ., 33(23), 3831-3842, 1999.

Rubin, J. I., Kean, A. J., Harley, R. A., Millet, D. B., and Goldstein, A. H.: Temperature dependence of volatile organic compound evaporative emissions from motor vehicles, J. Geophys. Res. Atmos., 111(D3), 2006.

Singer, B. C., Kirchstetter, T. W., Harley, R. A., Kendall, G. R., and Hesson, J. M.: A fuel-based approach to estimating motor vehicle cold-start emissions, J. Air Waste Manage. Assoc., 49(2), 125-135, 1999.

Reiter, M. S., and Kockelman, K. M.: The problem of cold starts: A closer look at mobile source emissions levels, Transp. Res. D: Transp. Environ., 43, 123-132, 2016.

Chiang, H. L., Tsai, J. H., Yao, Y. C., and Ho, W. Y.: Deterioration of gasoline vehicle emissions and effectiveness of tune-up for high-polluted vehicles, Transp. Res. D: Transp. Environ., 13(1), 47-53, 2008.

Borken-Kleefeld, J., and Chen, Y.: New emission deterioration rates for gasoline cars–Results from long-term measurements, Atmos. Environ., 101, 58-64, 2015.

5. Line 365: I think a Table can be added in SI to show the waterborne paint proportions within different application purposes. This would be helpful to other research on solvent use emissions.

**Response**: We thank the Referee for the constructive suggestion. A supplementary Table (i.e., Table S12) has been added in SI to show the waterborne paint proportions within each application purpose in 1970, 1990, 2010, and 2020.

6. There are a few minor grammatical issues. For instance, "besting performing" in Line 194 appears to be a typo. I recommend the authors carefully proofread the manuscript to correct such errors.

**Response**: We thank the Referee for the constructive comment. We have carefully proofread the manuscript and corrected the grammatical errors.

**Results**

1. Line 443: Abbreviations such as "PVC" should be spelled out in full when they first appear in the text, as not all readers, particularly those without a background in the chemical industry, may be familiar with such terms.

**Response**: We thank the Referee for the helpful comment. The full names have been provided in the manuscript as shown below.

*"Production of a few types of plastics—polyvinyl chloride (PVC), polypropylene (PP), polystyrene (PS), high-density polyethylene (HDPE), and low-density polyethylene (LDPE)— accounted for a large portion of NMVOC emissions (i.e., 58% and 76% of emissions from organic chemicals in 1970 and 2020)."*

2. Line 448: I think "mitigation priorities" would be better.

**Response**: We thank the Referee for the constructive suggestion. The sentence has been revised as suggested.

3. Line 465-485: Add some references to support the analysis, especially when explaining the reasons of emission changes, as some information is not directly reflected in the work.

**Response**: We thank the Referee for the constructive suggestion. The relevant references have been added to support the analysis of emission changes.

References:

Geng, G., Liu, Y., Liu, Y., Liu, S., Cheng, J., Yan, L., Wu, N., Hu, H., Tong, D. Zheng, B., Yin, Z., He, K., and Zhang, Q.: Efficacy of China's clean air actions to tackle $PM_{2.5}$ pollution between 2013 and 2020, Nat. Geosci., 17(10), 987-994, 2024.

Baidya, S., and Borken-Kleefeld, J.: Atmospheric emissions from road transportation in India, Energy Policy, 37(10), 3812-3822, 2009.

Saundry, P. D.: Review of the United States energy system in transition, Energy Sustain. Soc., 9(1), 4, 2019.

Pandey, A., and Venkataraman, C.: Estimating emissions from the Indian transport sector with on-road fleet composition and traffic volume, Atmos. Environ., 98, 123-133, 2014.

McDonald, B. C., Goldstein, A. H., and Harley, R. A.: Long-term trends in California mobile source emissions and ambient concentrations of black carbon and organic aerosol, Environ. Sci. Technol., 49(8), 5178-5188, 2015.

Plant, G., Kort, E. A., Brandt, A. R., Chen, Y., Fordice, G., Gorchov Negron, A. M., Schwietzke, S., Smith, M., and Zavala-Araiza, D.: Inefficient and unlit natural gas flares both emit large quantities of methane, Science, 377(6614), 1566-1571, 2022.

Tran, H., Polka, E., Buonocore, J. J., Roy, A., Trask, B., Hull, H., and Arunachalam, S.: Air quality and health impacts of onshore oil and gas flaring and venting activities estimated using refined satellite-based emissions, GeoHealth, 8(3), e2023GH000938, 2024.

Wu, W., Fu, T. M., Arnold, S. R., Spracklen, D. V., Zhang, A., Tao, W., Wang, X., Hou, Y., Mo, J., Chen, J., Li, Y., Feng, X., Lin, H., Huang, Z., Zheng, J., Shen, H., Zhu, L., Wang, C., Ye, J., and Yang, X. Temperature-dependent evaporative anthropogenic VOC emissions significantly exacerbate regional ozone pollution, Environ. Sci. Technol., 58(12), 5430-5441, 2024.

Klimont, Z., Amann, M., & Cofala, J.: Estimating costs for controlling emissions of volatile organic compounds (VOC) from stationary sources in Europe, https://pure.iiasa.ac.at/id/eprint/6195/, 2000.

4. I think "biofuel" would be better than "biomass" in the context.

**Response**: We thank the Referee for the constructive suggestion. The word "biomass" has been replaced with "biofuel" in the manuscript.

5. Line 562-563: While the authors discuss the case of India, similar emission characteristics and challenges are also prevalent in other developing regions, such as Africa. It is recommended that the authors add some short discussion to better reflect the urgent need for emission mitigation efforts across these regions.

**Response**: We thank the Referee for the thoughtful suggestion. A short discussion has been added as below.

*"Similar patterns were observed in other developing regions, such as Africa and Latin America, reflecting the urgent need to initiate and enhance NMVOC emission control through targeted policy implementation and technological advancement."*

**Discussions**

1. Line 592: Some research also claimed that agriculture could be an important source of NMVOC emissions (e.g., Hobbs et al., 2004). Some discussions would be appreciated.

Hobbs, P. J., Webb, J., Mottram, T. T., Grant, B., & Misselbrook, T. M. (2004). Emissions of volatile organic compounds originating from UK livestock agriculture. *Journal of the Science of Food and Agriculture*, *84*(11), 1414-1420.

**Response**: We thank the Referee for the constructive suggestion. The discussion has been revised as below.

*"A few minor sources, such as waste treatment and agriculture (Sharma et al., 2019; Hobbs et al., 2004), are also not included."*

2. Some activity rates are filled or modeled by statistical approaches, which introduces uncertainties. This should be mentioned in the limitation part.

**Response**: We thank the Referee for the helpful comment. The discussion on the limitation of activity rates has been added as below.

*"Secondly, part of the activity rates are reconstructed by statistical models, which introduces additional uncertainties and should be better constrained as higher-quality data are available in the future."*

3. In this work, NMVOC emissions are not speciated. This should be discussed and expected to be an important step in the future.

**Response**: We thank the Referee for the helpful comment. The discussion on NMVOC emission speciation has been added as below.

*"Finally, NMVOC emission speciation is not included in this study, but will be addressed in future work to provide species-resolved emissions and model-ready emissions for different chemical mechanisms."*